# The deubiquitinase USP17LA negatively regulates T-cell activation and attenuates anti-tumor immunity

Huiling Zhang [1,3], Zhihan Guo [1,3], Gaigai Wei [2], Jingjing Yi[2], Zixi Wang[2], Yuqi Zhang[1], Haiping Zhao [1], Tingrong Ren [1], Yihan Wang[1], Jiating Kuang [1], Zhaoying Sheng[1] & Duanwu Zhang [1✉]

## Abstract

T-cell activation is essential for effective immune responses, yet its precise regulatory mechanisms remain incompletely understood. In this study, we show that the deubiquitinases of the Ubiquitin-Specific Peptidase 17-like (USP17L) family are significantly upregulated following T-cell stimulation. Using CRISPR-mediated gene knockout mice, we demonstrate that USP17LA, but not USP17LB, acts as a negative regulator of T-cell activation. Loss of *Usp17la* leads to increased production of pro-inflammatory cytokines, enhanced T-cell proliferation and effector functions, without affecting T-cell development or homeostasis. Furthermore, *Usp17la* deletion augments TCR signaling and anti-tumor immunity, improving T-cell-mediated tumor surveillance in murine tumor models. Mechanistically, proteomic analysis revealed that USP17LA strongly associates with cadherin-binding and calmodulin-binding pathways. Notably, USP17LA interacts with RACK1 and prevents its ubiquitin-dependent degradation, thereby promoting RACK1-mediated suppression of NFAT activity and the subsequent inhibition of T-cell function. These findings establish USP17LA as a pivotal modulator of T-cell activation and suggest that targeting USP17LA could enhance anti-tumor immunity, offering a potential strategy for cancer immunotherapy.

**Keywords** USP17LA; T-cell Activation; Anti-tumor Immunity; NFAT; RACK1
**Subject Categories** Cancer; Immunology; Post-translational Modifications & Proteolysis

## Introduction

T-cell activation is a fundamental process in the immune system, playing a pivotal role in maintaining immune homeostasis and defending against infections and malignancies (Kishton et al, 2017).

Upon encountering antigens, T cells undergo a series of signaling events that lead to their proliferation, differentiation, and effector functions (Sun et al, 2023). Precise regulation of T-cell activation is essential, as dysregulation can result in immune dysfunction, including autoimmunity, immunodeficiency, or impaired anti-tumor immunity. Although many molecular components involved in T-cell activation have been identified (Chapman et al, 2020), the precise mechanisms underlying this process remain incompletely understood. Further investigation is required to identify additional regulatory factors that govern T-cell activation.

Deubiquitinating enzymes (DUBs) remove the Ub/UBL from modified proteins, reversing the effects of ubiquitin modifications (Patel et al, 2023a). Dysregulation of DUBs disrupts the dynamic equilibrium of the ubiquitome and causes various diseases, especially cancer and immune disorders (Li and Reverter, 2021). Among them, the Ubiquitin Specific Proteinase (USP) family is the largest subfamily of deubiquitinases, and its members have been widely reported to participate in T-cell activation (Ren et al, 2023). USP15 has been reported to inhibit anti-tumor T-cell responses through stabilizing MDM2 (Zou et al, 2014). USP8 has been reported to be critical for T-cell development and homeostasis by securing maturation and Foxo1-mediated upregulation of IL-7Rα (Dufner et al, 2015). Besides, A20, OTUB1, and OTUD2B were reported to be associated with T-cell activation (Düwel et al, 2009; Hu et al, 2016; Zhou et al, 2019). In this context, we noted that the USP17L family, which acts as a poor survival predictor in GVHD patients (Patel et al, 2023b) and is elevated in EL4 cells upon stimulation with PMA plus ionomycin, has not been studied in T cells.

Calcium signaling is essential for T-cell activation, with calmodulin (CaM) acting as the primary calcium sensor that orchestrates downstream effectors (Trebak and Kinet, 2019). Following T-cell receptor (TCR) engagement, calcium influx induces conformational changes in CaM, facilitating its interaction with key targets such as calcineurin and CaMKII. Calcineurin dephosphorylates nuclear factor of activated T cells (NFAT), promoting its nuclear translocation and cytokine transcription (e.g., IL-2, IFN-γ) (Macian, 2005). While calcium-related regulators remain to be fully elucidated, RACK1 knockdown has been

[1]Children's Hospital of Fudan University, National Children's Medical Center, and Shanghai Key Laboratory of Medical Epigenetics, International Co-laboratory of Medical Epigenetics and Metabolism, Ministry of Science and Technology, Institutes of Biomedical Sciences, Fudan University, 200032 Shanghai, China. [2]Institute of Pediatrics, Children's Hospital of Fudan University, National Children's Medical Center, Fudan University, 201102 Shanghai, China. [3]These authors contributed equally: Huiling Zhang, Zhihan Guo. ✉E-mail: duanwu@fudan.edu.cn

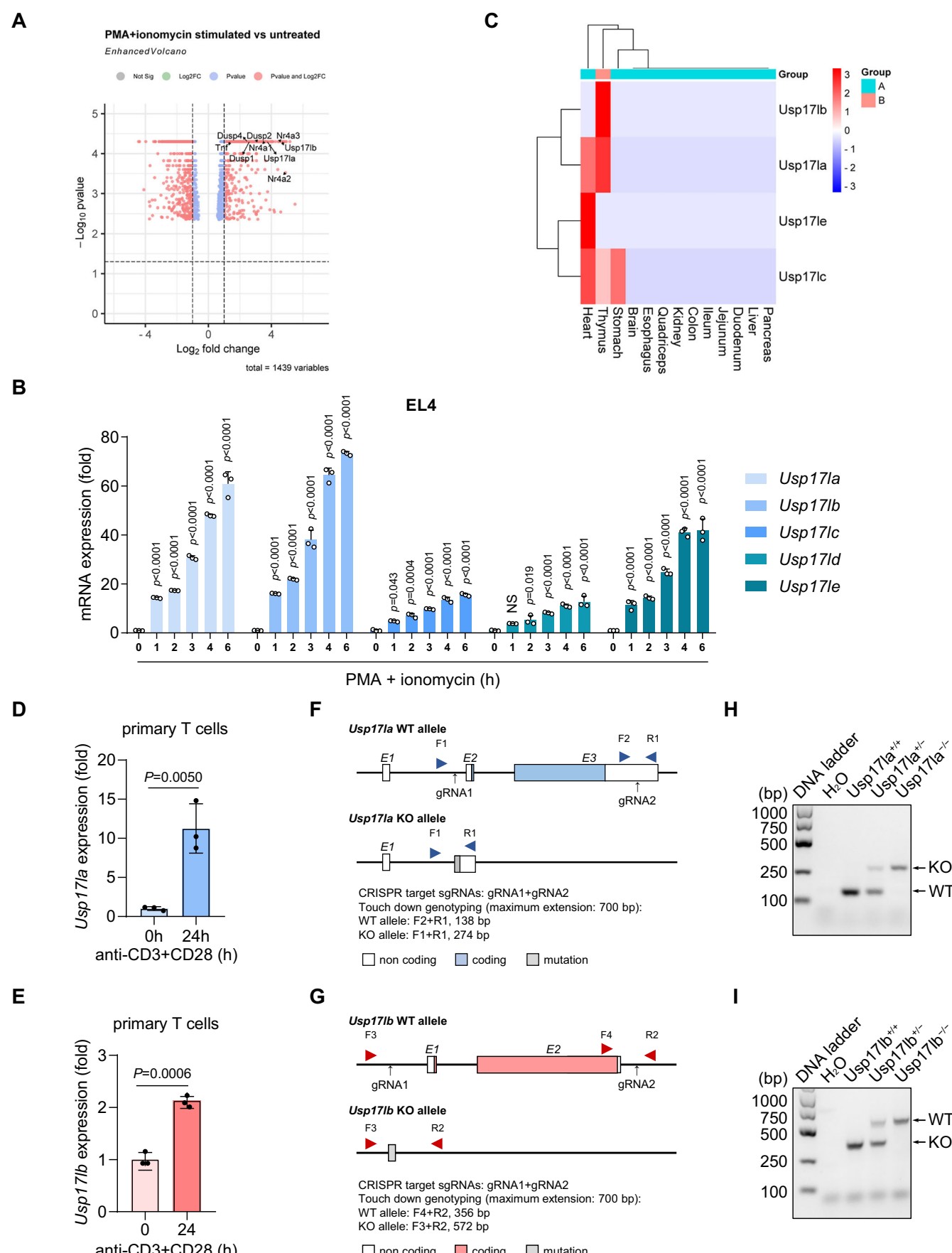

**Figure 1. USP17LA and USP17LB are potential regulatory factors for T-cell activation.**

(A) Volcano plot analysis of differentially expressed genes in PMA plus ionomycin stimulated EL4 cells versus untreated EL4 cells by RNA-seq analysis. The x axis represents $\log_2$ fold change of expression of these genes in the PMA plus ionomycin group compared to the untreated group, and the y axis represents $-\log_{10}$ P value. Data were obtained from our previous study (Zhang et al, 2019) and re-analyzed using the online bioinformatic platform (https://www.bioinformatics.com.cn). Differential expression was calculated with Cuffdiff v2.1.1 (n = 2 per group). (B) RT-qPCR analysis of the mRNA levels of *Usp17la*, *Usp17lb*, *Usp17lc*, *Usp17ld*, *Usp17le* in EL4 cells upon stimulation with PMA plus ionomycin for the indicated times, quantification was determined by using $2^{-\Delta\Delta Ct}$ methods (n = 3 per group). (C) Clustering analysis heatmap of *Usp17l*-family expression across various mouse tissues, with the immune-related tissues highlighted (pink group). Data were sourced from a previous study (Söllner et al, 2017) and analyzed using the same bioinformatic platform. (D, E) RT-qPCR analysis of *Usp17la* and *Usp17lb* mRNA levels in primary T cells upon stimulation with anti-CD3 and anti-CD28 for 24 h or not (n = 3, respectively). (F, G) Schematic diagram of the CRISPR/Cas9-mediated gene targeting strategy to generate *Usp17la*$^{-/-}$ and *Usp17lb*$^{-/-}$ mice, including the genotyping strategy for confirming targeted deletion. (H) Representative PCR genotyping results of *Usp17la*$^{+/+}$, *Usp17la*$^{+/-}$, and *Usp17la*$^{-/-}$ alleles using genomic DNA isolated from tail biopsies. (I) Representative PCR genotyping results of *Usp17lb*$^{+/+}$, *Usp17lb*$^{+/-}$, and *Usp17lb*$^{-/-}$ alleles using genomic DNA isolated from tail biopsies. Data are representative of three independent experiments (B, D, E, H, I). Data are shown as mean ± s.d. Statistical analyses were performed using two-way ANOVA followed by Dunnett's multiple comparisons test (B) and unpaired, two-tailed Student's t test (D, E). NS not significant. Source data are available online for this figure.

reported to enhance antigen-induced calcium mobilization in mast cells (Filho et al, 2021). Moreover, RACK1 inhibits NFAT activity (Han et al, 2002), suggesting that its regulation modulates the calcium-NFAT signaling axis.

Here, we utilized *Usp17la*-deficient mice to demonstrate that *Usp17la* deficiency does not affect T-cell development in the thymus or peripheral lymphoid organs. However, loss of *Usp17la* specifically enhances T-cell activation by promoting TCR signaling. Mechanistically, USP17LA interacts with calcium-related pathway proteins and facilitates NFAT activation by mediating the ubiquitin-dependent degradation of RACK1. RACK1 overexpression inhibits excessive cytokine production caused by USP17LA depletion. Furthermore, *Usp17la* deficiency suppresses tumor growth and enhances T-cell anti-tumor efficacy. These findings reveal a previously unrecognized role of USP17LA as a negative regulator of T-cell activation.

## Results

### USP17-like deubiquitinases are significantly induced during T-cell activation

To identify new regulators involved in T-cell activation, we analyzed our previous RNA sequencing data from PMA and ionomycin-stimulated EL4 cells (Zhang et al, 2019). RNA-sequencing analysis revealed 982 differentially expressed genes (DEGs), including 536 upregulated and 446 downregulated genes (Fig. 1A). Among the upregulated genes, negative regulators of TCR signaling, such as *Nr4a1*, *Nr4a3*, *Dusp2*, and *Dusp6*, were significantly increased (Bertin et al, 2015; Chen et al, 2019; Dan et al, 2020; Hsu et al, 2018; Liu et al, 2019). Notably, members of the deubiquitinating enzyme (DUB) family, including *Usp17la*, *Usp17lb*, *Usp17ld*, and *Usp17le*, were also upregulated (Fig. 1A), suggesting their potential role in modulating T-cell activation. Gene ontology (GO) enrichment and KEGG pathway analyses of the upregulated DEGs highlighted pathways related to protein phosphorylation, cell adhesion, catabolic processes, and immune signaling, such as IL-17, NF-κB, and MAPK pathways, among the top 20 enriched pathways (Fig. EV1A,B).

Given the critical role of DUBs in regulating protein function, the *Usp17*-like family members may provide mechanistic insights into immune modulation. Then, we validated the expression

profiles of the identified genes in EL4 cells following PMA and ionomycin stimulation. Compared to control cells, the mRNA levels of all identified genes were significantly increased in a time-dependent manner (Fig. 1B). Tissue expression analysis further revealed that these DUBs were highly enriched in the thymus and heart (Fig. 1C), implicating them in T-cell development, activation, and heart homeostasis. Due to their high sequence similarity (Fig. EV1C), USP17L family members may be functionally redundant and complex. Based on their pronounced expression in the thymus and in PMA-stimulated EL4 cells, *Usp17la* and *Usp17lb* were selected for further investigation. Their upregulation was confirmed in primary T cells stimulated with anti-CD3 and anti-CD28 antibodies (Fig. 1D,E).

Then, we generated *Usp17la*$^{-/-}$ and *Usp17lb*$^{-/-}$ mice using the CRISPR/Cas9 strategy to facilitate further investigation (Fig. 1F,G). Genotypes of these knockout mice were confirmed by tail DNA genotyping (Fig. 1H,I) and RT-PCR of spleen T-cell mRNA (Fig. EV1D–G), validating gene deletion at both DNA and mRNA levels. Due to the high amino acid sequence similarity among USP17L family proteins, we were unable to generate USP17LA- and USP17LB-specific antibodies, which prevented us from detecting the effects of USP17LA and USP17LB deletion in mice at the protein level. The mice were born in accordance with Mendelian inheritance laws and exhibited no significant differences in weight or appearance compared to their wild-type (WT) counterparts (Fig. EV1H–J). In summary, USP17LA and USP17LB are identified as DUBs upregulated during T-cell activation, suggesting their roles in immune regulation.

### *Usp17la* deficiency does not affect T-cell homeostasis or development

We then examined whether *Usp17la* or *Usp17lb* deficiency affects T-cell homeostasis. Ablation of *Usp17la* did not alter the size or cellularity of inguinal lymph nodes (Fig. 2A,B). Detailed analysis of immune cell subpopulations in lymph nodes revealed no significant differences in the absolute numbers or proportions of B cells, CD3$^+$ T cells, CD4$^+$ T cells, CD8$^+$ T cells, macrophages, NK cells, or NKT cells in *Usp17la*$^{-/-}$ mice compared to WT controls (Figs. 2C,D and EV2A). Similarly, the size, total cellularity, and composition of immune subsets in the spleen were comparable between genotypes (Figs. 2E–H and EV2B). Regulatory T-cell (Treg) frequencies in both spleen and lymph nodes were also unaffected (Fig. 2I,J). Along with

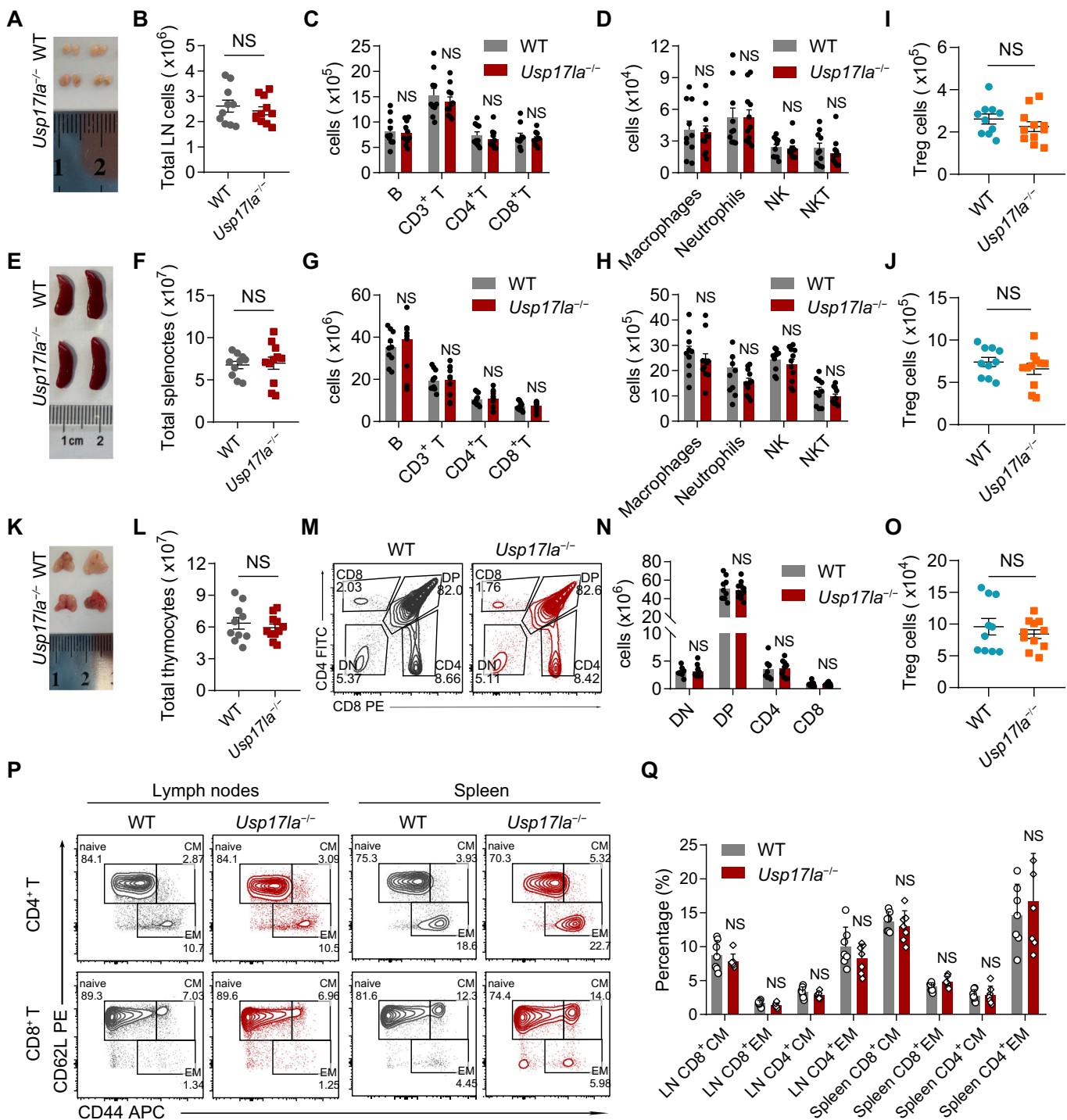

**Figure 2. *Usp17la* ablation does not affect peripheral T-cell homeostasis or thymic T-cell development.**

(A–D) Inguinal lymph nodes from 2-month-old WT and *Usp17la⁻/⁻* mice were isolated and analyzed by organ size (A), organ cell numbers (B), and absolute number of various immune cell types (C, D) ($n = 10$–11 per genotype). (E–H) Spleens from 2-month-old WT and *Usp17la⁻/⁻* mice were isolated and analyzed by organ size (E), organ cell numbers (F), and absolute number of various immune cell types (G, H) ($n = 10$–11 per genotype). (I, J) Flow cytometric quantification of Foxp3⁺ CD4⁺ Treg cells in the lymph nodes and spleen of WT and *Usp17la⁻/⁻* mice ($n = 10$–11 per genotype). (K–O) Thymuses from 2-month-old WT and *Usp17la⁻/⁻* mice were isolated and analyzed by organ size (K), organ cell numbers (L), representative flow cytometry plots (M), absolute number of DN, DP, CD4 SP, CD8 SP subpopulations (N), and absolute number of Treg cells (O) ($n = 10$–11 per genotype). (P, Q) Flow cytometry analysis of the expression of CD44 and CD62L in CD4⁺ and CD8⁺ T cells from lymph nodes and spleens of WT and *Usp17la⁻/⁻* mice ($n = 7$ per genotype). Representative flow cytometry plots (P) and quantification (Q) are shown. Data are a combination of two independent experiments (A–Q). Data are shown as mean ± s.d. Statistical analyses were performed using unpaired, two-tailed Student's *t* test (B, F, I, J, L, O) and two-way ANOVA followed by Sidak's multiple comparisons test (C, D, G, H, N, Q). NS not significant. Source data are available online for this figure.

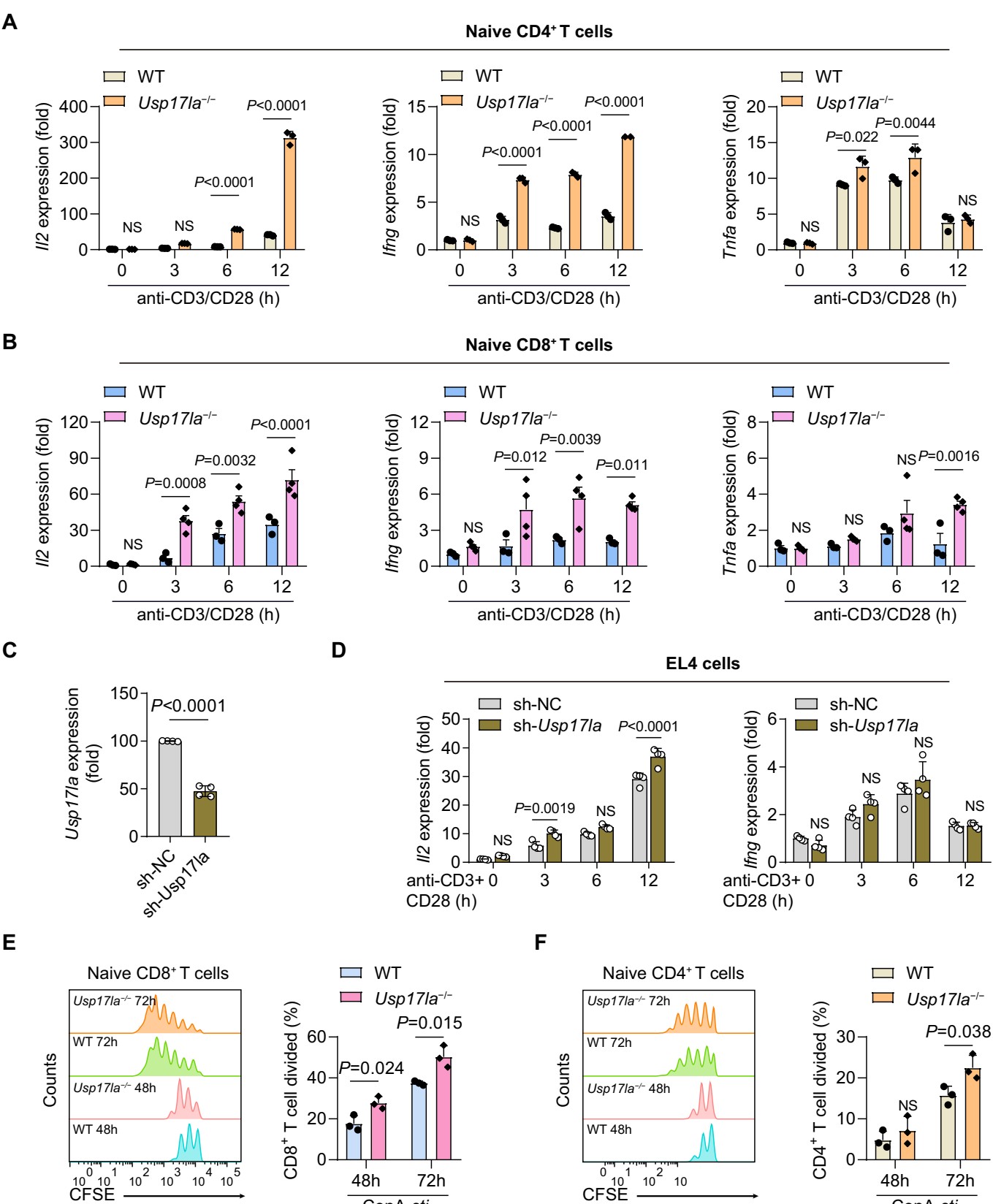

◀  **Figure 3.  *Usp17la* deletion promotes activation and expansion of T cells.**

(A) RT-qPCR analysis of the expression of *Il-2, Ifng*, and *Tnfa* in WT and *Usp17la*$^{-/-}$ CD4$^+$ T cells upon stimulation with anti-CD3 plus anti-CD28 for the indicated times ($n = 3$ per genotype). (B) RT-qPCR analysis of the expression of *Il-2, Ifng*, and *Tnfa* in WT and *Usp17la*$^{-/-}$ CD8$^+$ T cells upon stimulation with anti-CD3 plus anti-CD28 for the indicated times ($n = 3$–4 per genotype). (C) RT-PCR analysis of *Usp17la* knockdown efficiency in EL4 cells following infection with sh-NC or sh-*Usp17la* lentivirus ($n = 4$ per group). (D) RT-PCR analysis of the expression of *Il-2* and *Ifng* in EL4 cells infected with sh-NC or sh-*Usp17la* lentivirus following anti-CD3 plus anti-CD28 stimulation for the indicated times ($n = 4$ per group). (E, F) Flow cytometry analysis of the proliferation of naive CD8$^+$ and CD4$^+$ T cells purified from the spleen of WT and *Usp17la*$^{-/-}$ mice following stimulation with ConA for the indicated time periods. The cell division percent was measured by CFSE dilution ($n = 3$ per genotype). Data are representative of two independent experiments (A–F). Data are shown as mean ± s.d. Statistical analyses were performed using two-way ANOVA followed by Sidak's multiple comparisons test (A, B, D) and unpaired, two-tailed Student's *t* test (C, E, F). NS not significant. Source data are available online for this figure.

unchanged T-cell number in the periphery, T-cell development in the thymus remained intact in *Usp17la*$^{-/-}$ mice, as evidenced by normal thymic morphology, thymocyte counts, and thymocyte subpopulations based on CD4 and CD8 expression (Figs. 2K–N and EV2C). Further analysis of early thymocyte development revealed no significant differences in DN subpopulations (DN1–DN4), defined by CD44 and CD25 expression, between *Usp17la*$^{-/-}$ and WT mice (Fig. EV2D–F). Moreover, Treg development in the thymus was also unaffected in *Usp17la*$^{-/-}$ mice (Fig. 2O).

We also assessed the activation status of peripheral T cells by analyzing CD44 and CD62L expression, which allowed for the classification of T cells into naive, central memory, and effector memory subpopulations. No significant differences were observed between *Usp17la*$^{-/-}$ and WT mice (Fig. 2P,Q). Overall, these findings indicate that *Usp17la* deficiency does not significantly affect T-cell development or peripheral homeostasis.

Similarly, *Usp17lb* ablation had no effect on T-cell percentages or the CD4$^+$/CD8$^+$ ratio in lymph nodes or spleen (Fig. EV3A–D). In contrast to the unchanged activation status observed in *Usp17la*$^{-/-}$ mice, CD8$^+$ T cells in *Usp17lb*$^{-/-}$ mice exhibited a decreased proportion of central memory cells in both the lymph nodes and spleen (Fig. EV3E,F). Analysis of thymocyte subpopulations, defined by CD4 and CD8 expression, and DN subpopulations, defined by CD44 and CD25, revealed no significant differences between *Usp17lb*$^{-/-}$ and WT mice, indicating no apparent changes in thymocyte development (Fig. EV3G,H). In summary, *Usp17lb* deficiency does not appear to have any significant effect on T-cell development and homeostasis.

## USP17LA restrains T-cell activation and expansion

The induced expression of *Usp17la* and *Usp17lb* in primary T cells following TCR engagement suggested their involvement in T-cell activation. To assess the role of USP17LA in this process, we performed in vitro activation studies using anti-CD3 and anti-CD28 antibodies. Our results revealed that *Usp17la*$^{-/-}$ naive CD4$^+$ T cells exhibited significantly higher production of *IL-2, IFN-γ*, and *TNF-α* at various time points following stimulation (Fig. 3A). Similarly, *Usp17la*$^{-/-}$ naive CD8$^+$ T cells exhibited augmented effector function, as evidenced by increased production of *IL-2, IFN-γ, and TNF-α* (Fig. 3B). In EL4 cells, *Usp17la* knockdown upregulated *IL-2* and *IFN-γ* levels upon TCR engagement (Fig. 3C,D), while *Usp17lb* knockdown had the opposite effect when stimulated for 12 h (Fig. EV3I,J). These findings suggest that USP17LA negatively regulates the activity of both CD4$^+$ and CD8$^+$ T cells.

Consistent with these observations, *Usp17la*$^{-/-}$ CD4$^+$ and CD8$^+$ T cells showed significantly higher proliferation in response to Concanavalin A (ConA) stimulation, with the percentage of divided cells increasing by ~50% (Fig. 3E,F). In summary, USP17LA acts as a negative regulator of T-cell activation and expansion by restraining effector function and proliferation in both CD4$^+$ and CD8$^+$ T cells.

## *Usp17la* deficiency enhances T-cell-mediated anti-tumor immunity

Given that *Usp17la* deficiency enhanced T-cell activity, it was critical to explore its physiological implications under pathological conditions. To this end, we first utilized the MC38 murine colon cancer model to evaluate the anti-tumor effects of *Usp17la*$^{-/-}$ mice. Compared to WT controls, *Usp17la*$^{-/-}$ mice exhibited significantly reduced tumor progression and smaller tumor sizes (Figs. 4A–D and EV4A–C), suggesting that USP17LA functions as a negative regulator of anti-tumor responses. Further analysis of tumor-infiltrating lymphocytes (TILs) isolated from MC38 tumors revealed that *Usp17la* deficiency increased granzyme B (GZMB) and IFN-γ production by cytotoxic CD8$^+$ T cells (Fig. 4E,F). Moreover, the frequency of Treg cells in tumor-draining lymph nodes and spleen remained unchanged in *Usp17la*$^{-/-}$ mice (Fig. 4G). These data indicate that *Usp17la* deficiency can enhance anti-tumor immunity against MC38 solid tumors.

We also examined the role of USP17LA in metastatic tumor progression using the B16/F10 melanoma lung metastasis model. *Usp17la*$^{-/-}$ mice exhibited less severe tumor invasion compared to WT mice (Fig. 4H,I). To further explore the effect of USP17LA on antigen-specific CD8$^+$ T cell-mediated anti-tumor immunity, we utilized OT–I transgenic mice. Co-culture of OVA$_{257-264}$ peptide-stimulated OT–I CD8$^+$ T cells with OVA-expressing B16/F10 melanoma cells demonstrated that *Usp17la*$^{-/-}$ OT–I CD8$^+$ T cells exhibited enhanced cytotoxicity, as indicated by increased lactate dehydrogenase release (Fig. 4J). These results demonstrate that *Usp17la* deficiency enhances anti-tumor immunity by promoting CD8$^+$ T-cell effector functions. We next evaluated the susceptibility of *Usp17la*-deficient mice to DSS-induced colitis and found that loss of *Usp17la* had no effect on body weight loss during the course of treatment (Fig. EV4D), suggesting that USP17LA is dispensable for innate immune responses to epithelial injury.

In contrast to the enhanced anti-tumor efficacy observed in *Usp17la*$^{-/-}$ mice, *Usp17lb*$^{-/-}$ mice showed accelerated tumor progression (Fig. EV4E–G), indicating a potential tumor suppressive role for USP17LB. Despite being homologous members of the

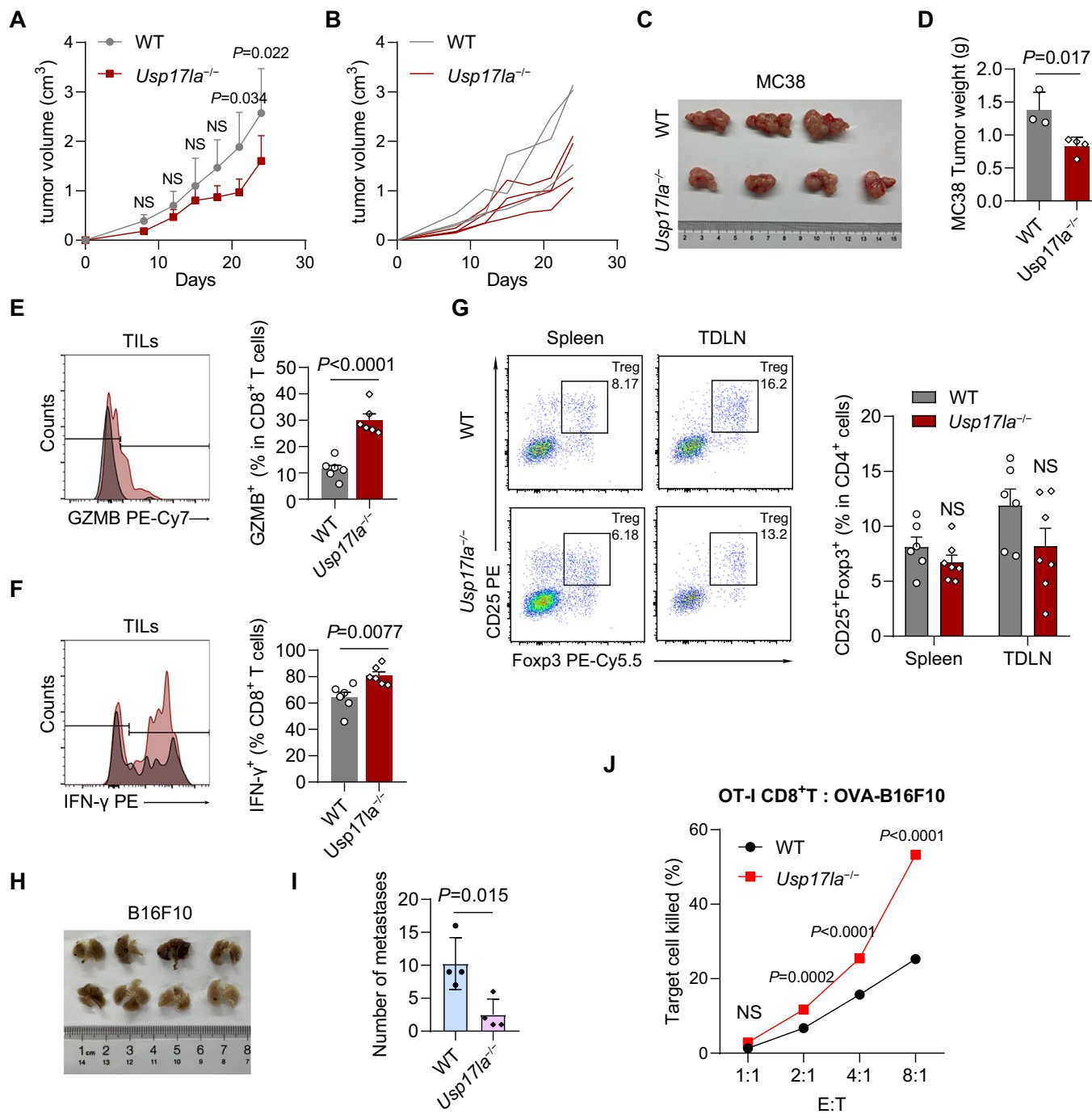

**Figure 4. *Usp17la* deficiency enhances T-cell anti-tumor immunity.**

(A–D) Tumor growth was assessed in terms of the tumor volume (A, B), tumor size (C), and tumor weight (D) in WT and *Usp17la$^{-/-}$* mice after MC38 colon cancer cells inoculation (WT, $n = 3$; *Usp17la$^{-/-}$*, $n = 4$). (E, F) Flow cytometry analysis of the expression of GZMB (D) and IFN-γ (E) in tumor-infiltrating CD8$^+$ T cells from MC38 tumor−bearing mice on day 25 ($n = 6$ per genotype). (G) Flow cytometry analysis of CD25$^+$ Foxp3$^+$ Treg cells in tumor-infiltrating CD4$^+$ T cells from MC38 tumor−bearing mice on day 25 ($n = 6$ per genotype). (H, I) Representative images of lung metastases and quantification of the total number of lung surface metastases from WT and *Usp17la$^{-/-}$* mice 14 days after tail-vein injection with B16F10 melanoma cells ($n = 4$ per genotype). (J) In vitro CTL cytotoxicity assay of OVA$_{257-264}$ peptide−pulsed B16/F10 cells co-cultured with WT OT − I and *Usp17la$^{-/-}$* OT − I CD8$^+$ T cells for 4 h at the indicated Effector:Target (E:T) ratio ($n = 3$ per group). Data are representative of two independent experiments (A–D, H–J) or pooled from two independent experiments (E–G). Data are shown as mean ± s.d. (J, error bars are present but not readily discernible due to minimal variance among replicates). Statistical analyses were performed using two-way ANOVA followed by Sidak's multiple comparisons test (A, G, J), and unpaired, two-tailed Student's *t* test (D-F, I). NS not significant. Source data are available online for this figure.

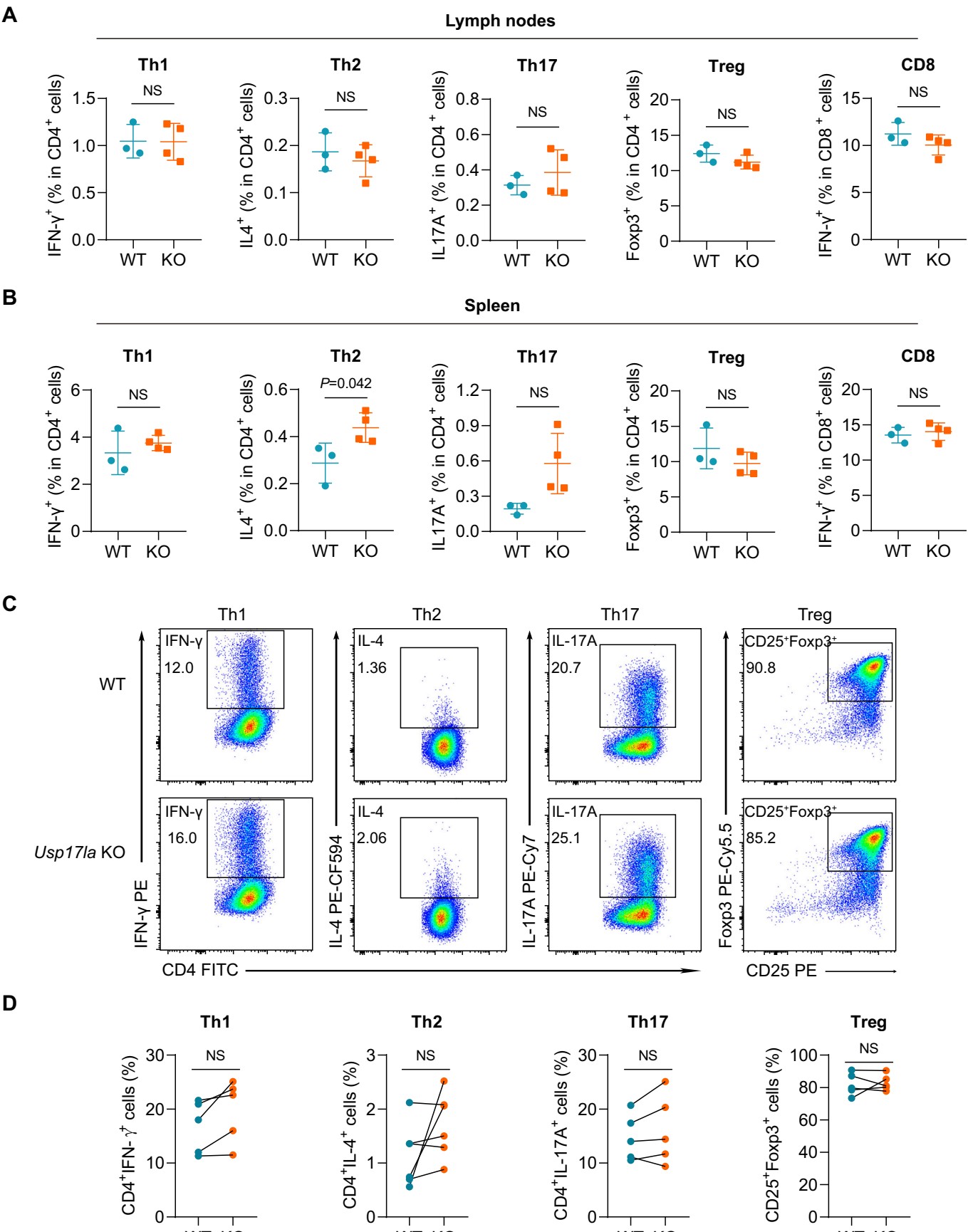

Figure 5.  *Usp17la* deficiency does not alter Th cell distribution or differentiation.

(A, B) Flow cytometry analysis of the expression of IFN-γ, IL-4, IL-17A, and Foxp3 in CD4$^+$ T cells and IFN-γ in CD8$^+$ T cells from lymph nodes and spleens of WT and *Usp17la$^{-/-}$* mice ($n = 3$–4 per genotype). (C, D) Flow cytometry analysis of the staining of IFN-γ, IL-4, IL-17A, CD25, and Foxp3 in WT and *Usp17la$^{-/-}$* naive CD4$^+$ T cells activated with plate-bound anti-CD3 plus anti-CD28 under various polarization conditions. Representative flow cytometry plots (C) and quantification (D) are shown. ($n = 5$–6 per genotype). Data are representative of two independent experiments (A–D). Data are shown as mean ± s.d. Statistical analyses were performed using unpaired, two-tailed Student's *t* test (A, B) and paired, two-tailed Student's *t* test (D). NS not significant. Source data are available online for this figure.

USP17L family, USP17LA and USP17LB exhibited distinct roles in tumor regulation.

## *Usp17la* is dispensable for the homeostasis and differentiation of CD4$^+$ T helper cells

In addition to investigating the effects of USP17LA on CD8$^+$ T-cell function, we further examined whether *Usp17la* deficiency impacts CD4$^+$ T-cell homeostasis. Th2 population was slightly increased in the spleens of *Usp17la$^{-/-}$* mice, but no significant differences in other Th populations (Th1, Th2, Th17, Treg) were observed in their lymph nodes (Fig. 5A,B). In addition, no differences were observed in IFN-γ$^+$ CD8$^+$ T cells between the two genotypes (Fig. 5A,B). Moreover, *Usp17lb* deficiency did not affect the proportion of Th1 and Th17 cells in the spleen under homeostasis (Fig. EV4H,I).

To evaluate whether *Usp17la* deficiency influences T-cell differentiation, we stimulated naive CD4$^+$ T cells from WT and *Usp17la*-deficient mice with anti-CD3 and anti-CD28 antibodies under Th1, Th2, Th17, and iTreg-polarizing conditions. Flow cytometric analysis revealed no significant differences in the expression of signature cytokines, including IFN-γ, IL-4, and IL-17A, between the two groups (Fig. 5C,D). Moreover, the induction efficiency of Foxp3$^+$ iTreg cells was comparable (Fig. 5C,D). These findings suggest that USP17LA does not significantly impact CD4$^+$ T-cell homeostasis or differentiation.

### Loss of *Usp17la* enhances TCR signaling

Signaling pathways, including MAPK, NF-κB, and NFAT, are integral to T-cell activation processes (Blanchett et al, 2021; Rincón et al, 2001). To investigate how *Usp17la* deletion affects T-cell activation, we analyzed intracellular signaling events in splenic primary T cells from WT and *Usp17la* KO mice following stimulation with anti-CD3 plus anti-CD28 antibodies or PMA plus ionomycin. Both stimuli activated downstream TCR signaling molecules, including p-ERK and p-P65 (Fig. 6A,B). In *Usp17la*-knockout T cells, phosphorylation of ERK and P65 was significantly elevated (Fig. 6A,B). This enhancement was also validated in *Usp17la* KO EL4 cells compared to the sg-Renilla controls (Fig. 6C,D). Conversely, USP17LB overexpression enhanced TCR signaling, as indicated by increased phosphorylation of ERK, JNK, P38, IKKα/β, and P65 (Fig. EV4J).

Reporter assays were employed to assess the effects of USP17LA and USP17LB on NFAT, NF-κB, and MAPK signaling pathways. In these assays, HEK293T cells were co-transfected with reporter plasmids and candidate plasmids, followed by stimulation with PMA plus ionomycin to measure luminescence reporter activity (Fig. 6E). The positive control, DUSP2, significantly inhibited NFAT reporter activity, consistent with a previous report (Lu et al,

2015). Similarly, USP17LA and USP17LB inhibited NFAT activity but did not affect NF-κB or MAPK activity (Fig. 6F). These findings suggest potential feedback mechanisms associated with USP17LA and USP17LB ablation and overexpression. Collectively, these findings indicate that USP17LA functions as a negative regulator of TCR signaling pathways.

## USP17LA modulates the Ubiquitin-mediated degradation of RACK1 to regulate NFAT signaling

To further elucidate the molecular mechanism regulated by USP17LA, we conducted immunoprecipitation coupled with mass spectrometry analysis (IP/MS) and identified 229 potential interactors of USP17LA. Tissue expression enrichment analysis using STRING revealed that these proteins are predominantly expressed in immune cells, consistent with their roles in T-cell activation (Fig. 7A). DAVID-based Gene Ontology molecular function (GO-MF) analysis identified cadherin binding and calmodulin binding as two of the top 10 enriched pathways (Fig. 7B). Cadherins and calmodulins are key molecules involved in T-cell activation. Cadherins play a critical role in the formation of the immunological synapse and signal transduction (Charnley et al, 2023). In contrast, calmodulin, as a central calcium signaling adapter, directly regulates the dephosphorylation and nuclear translocation of NFAT (Trebak and Kinet, 2019). The interactions between USP17LA and cadherin-binding proteins or calmodulin-binding proteins may alter the function of cadherins or calmodulin, thereby inhibiting downstream signaling in T cells.

Analysis of candidate USP17LA-interacting partners revealed that RACK1 (Fig. 7C), acting as a receptor for PKC, inhibits NFAT transcriptional activity (Han et al, 2002). Besides, previous study indicates that loss of RACK1 elicits pathological hyperactivation in CD8$^+$ T lymphocytes (Qiu et al, 2017), suggesting its putative role as a negative regulator of T-cell activation signaling pathways. First, we verified that USP17LA interacts with RACK1 in co-expression settings (Fig. 7D). Ubiquitination assays demonstrated that USP17LA suppresses K48-linked ubiquitination of RACK1, thereby attenuating its proteasomal degradation and stabilizing RACK1 protein levels (Fig. 7E). Reconstitution of RACK1 expression in USP17LA-depleted EL4 cells suppressed the upregulation of IL-2 and IFN-γ induced by anti-CD3 and anti-CD28 stimulation, supporting the notion that USP17LA regulates T-cell activation through a RACK1-dependent mechanism (Figs. 7F,G and EV5A). Consistently, RACK1 overexpression dampened the enhanced NFAT luciferase reporter activity observed in *Usp17la*-deficient cells under the same stimulatory conditions (Figs. 7H and EV5B). Collectively, these findings suggest that USP17LA regulates T-cell activation by modulating NFAT activity through stabilization of RACK1.

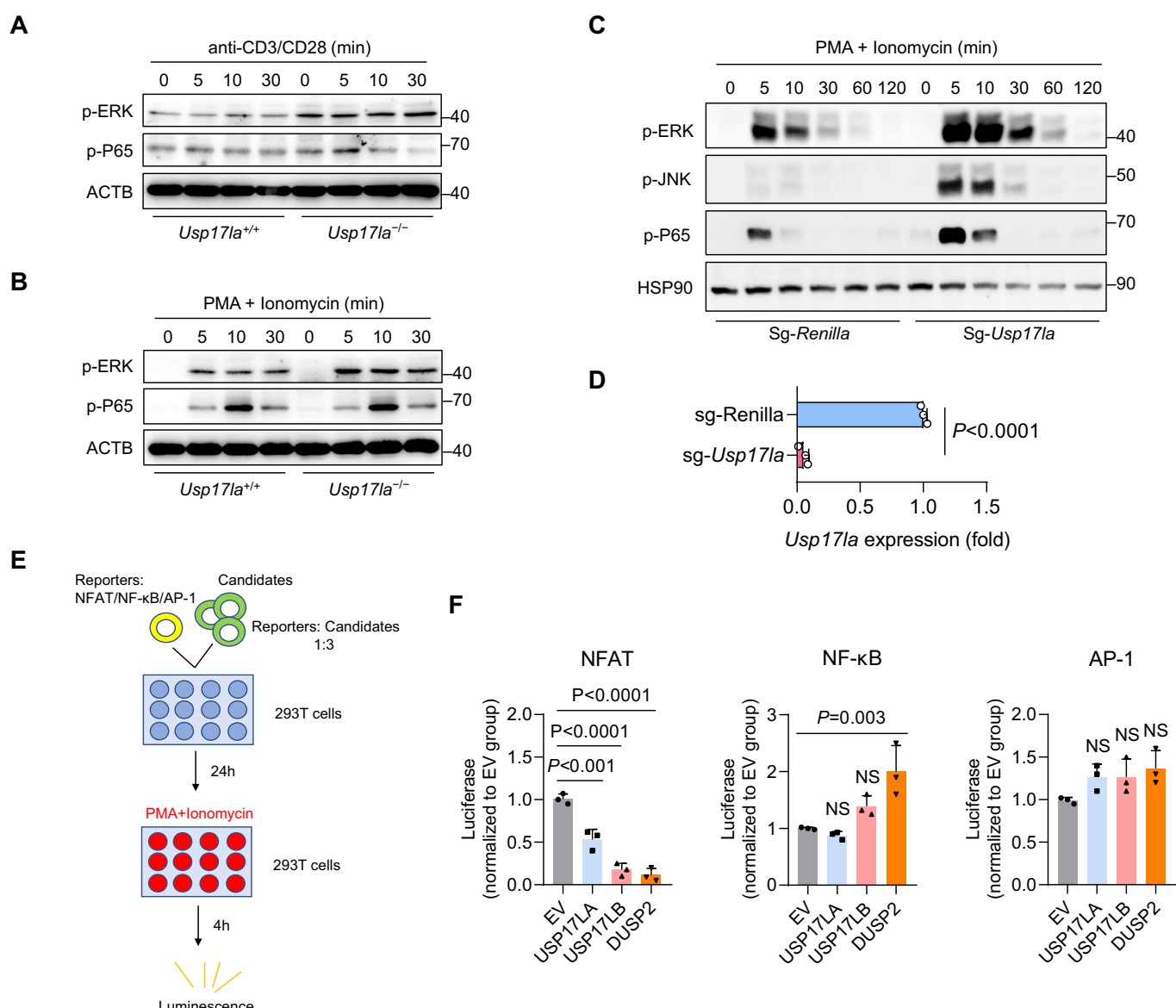

**Figure 6. USP17LA negatively modulates T-cell activation through inhibiting MAPK-NF-κB-NFAT signaling pathway.**

(A, B) Immunoblot analysis of TCR signaling-associated proteins in anti-CD3 plus anti-CD28-stimulated (A) and PMA plus ionomycin-stimulated (B) CD3+ T cells from WT and *Usp17la*−/− splenocytes. (C) Immunoblot analysis of TCR signaling-associated proteins in PMA plus ionomycin-stimulated EL4 cells transduced with a sgRNA control (sg-Renilla) or a *Usp17la*-specific sgRNA (sg-Usp17la). (D) RT-qPCR analysis of *Usp17la* mRNA levels in EL4 cells as in (C) (*n* = 3 per group). (E) Schematic image of transcription factor reporter assay. HEK293T cells were stimulated with PMA plus ionomycin after co-transfection with reporter plasmids and EV, USP17LA, or USP17LB for 24 h and then lysed to detect luminescence. (F) The relative luciferase reporter values of USP17LA, USP17LB, and DUSP2 were normalized to those of the EV group (*n* = 3 per group). Data are representative of two (A, B) or three (C–F) independent experiments. Data are shown as mean ± s.d. Statistical analyses were performed using unpaired, two-tailed Student's *t* test (D) and one-way ANOVA with Dunnett's multiple comparisons test (F). NS not significant. Source data are available online for this figure.

# Discussion

T-cell activation is fundamental to the adaptive immune response, playing a crucial role in defense against infections and immune homeostasis. While numerous studies have identified key regulators of T-cell activation, the precise molecular mechanisms remain incompletely understood. In this study, we identified USP17LA and USP17LB, two members of the deubiquitinase (DUB) family, as novel regulators of T-cell activation with opposing functions.

*Usp17la* deficiency enhanced T-cell activation without affecting T-cell development, whereas *Usp17lb* deficiency reduced T-cell activation. Notably, *Usp17la* knockout mice exhibited heightened anti-tumor immunity, suggesting a potential role for USP17LA in tumor immunosurveillance.

Mechanistically, USP17LA interacts with cadherin-binding and calmodulin-binding proteins and suppresses K48-linked ubiquitination of RACK1, thereby stabilizing NFAT activity and enhancing T-cell activation. USP17LA deletion also enhances ERK and P65

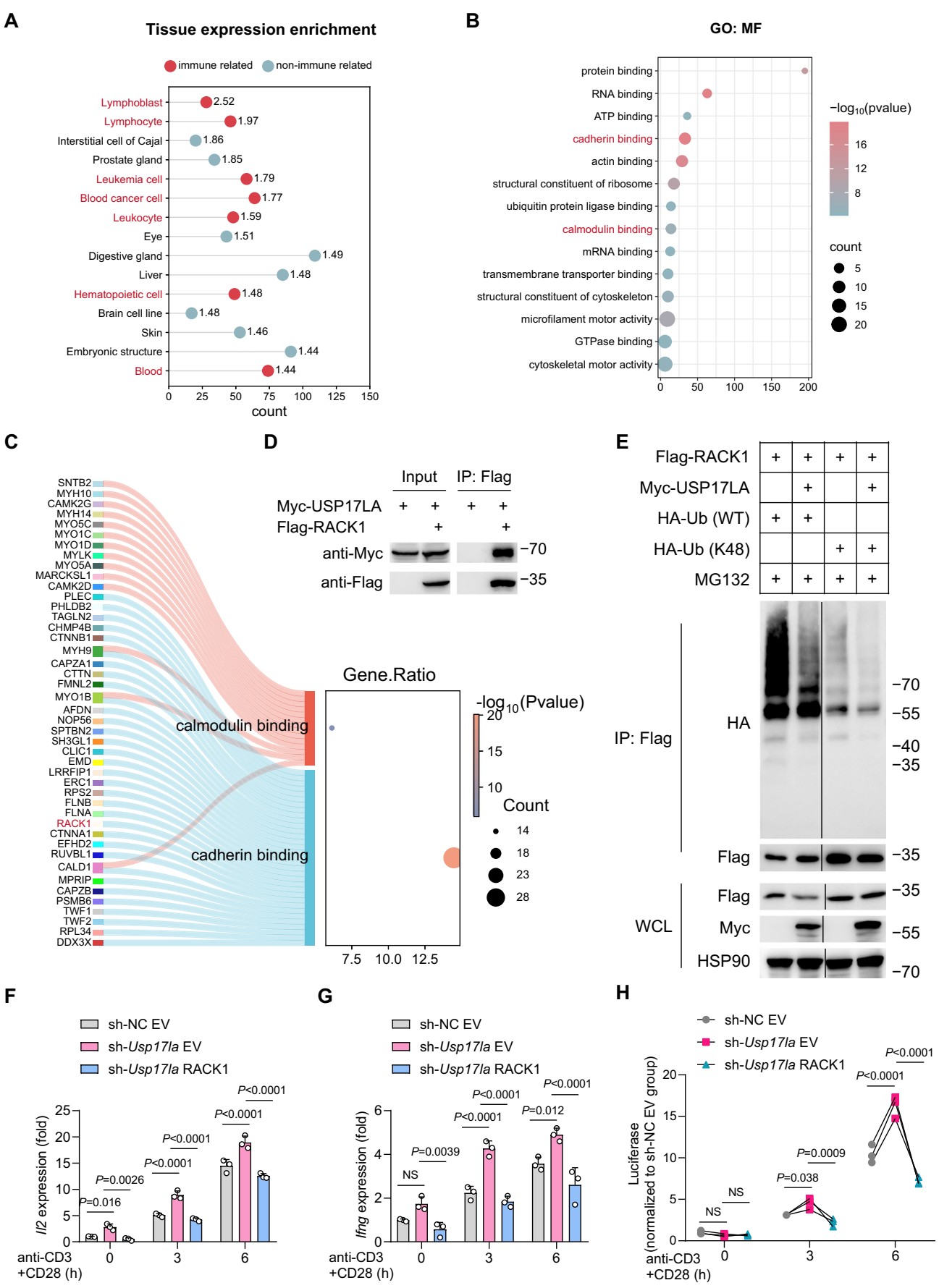

◀ **Figure 7.  USP17LA binds to cadherin-binding and calmodulin-binding proteins and inhibits NFAT activity by impairing RACK1 ubiquitination degradation.**

(A) Tissue expression profile of Flag-USP17LA-interacting proteins identified by IP/MS was analyzed using the STRING database (STRING: functional protein association networks). (B) GO molecular function analysis of USP17LA-immunoprecipitated proteins was performed using the DAVID database (https://david.ncifcrf.gov/), with a cutoff spectral index ratio of USP17LA versus vector set at 2. (C) Image representation of proteins involved in cadherin-binding and calmodulin-binding pathways enriched by Flag-USP17LA IP/MS. Image generated via a bioinformatic tool (www.microbioinformatics.cn). (D) Interaction was validated in HEK293T cells by co-expressing Flag-tagged RACK1 and Myc-tagged USP17LA, followed by immunoprecipitation using anti-Flag M2 beads and subsequent immunoblotting. (E) Immunoblot analysis of lysates from HEK293T cells co-transfected with Flag-RACK1, Myc-USP17LA, and HA-Ub (WT or K48 mutant), followed by a 4-h incubation with 20 mM MG132. (F, G) RT-PCR analysis of the expression of *Il-2* and *Ifng* in EL4 cells following infection with sh-NC or sh-*Usp17la* in combination with OE-EV or OE-RACK1 lentiviruses, following anti-CD3 plus anti-CD28 stimulation for the indicated times ($n = 3$ per group). (H) Relative luciferase activity in NFAT-luciferase reporter EL4 cells expressing sh-NC or sh-*Usp17la*, with or without RACK1 overexpression, following anti-CD3 plus anti-CD28 stimulation ($n = 3$ per group). Data are representative of two (D, E, H) or three (F, G) independent experiments. Statistical analyses were performed using two-way ANOVA with Turkey's multiple comparisons test (F, G) and with Dunnett's multiple comparisons test (H). NS not significant. Source data are available online for this figure.

phosphorylation in EL4 and primary T cells; however, AP-1 and NF-κB reporter activity in HEK293T cells remains unchanged. This discrepancy likely reflects differences in cellular context: 293T cells lack intact TCR signaling, limiting faithful AP-1/NF-κB readouts, whereas NFAT activity is preserved via the calcium–calcineurin pathway. Similarly, DUSP2, a known MAPK regulator in T cells (Dan et al, 2020), fails to suppress AP-1 in 293T cells, underscoring the limitations of this system. Therefore, these findings are complementary rather than contradictory. However, our conclusions regarding CD4⁺ T-cell activation in *Usp17la*-deficient cells remain cautious, as the depletion method used did not specifically exclude Treg cells, which may influence activation outcomes.

Our study showed that USP17LA can regulate the stability of RACK1, thereby affecting NFAT activity, but did not explain how USP17LA regulates MAPK and NFKB activity. Prior studies reported RACK1 negatively regulates NF-κB activity in gastric cancer cells (Yong-Zheng et al, 2015), potentially explaining increased NF-κB activation in *Usp17la*⁻/⁻ T cells. Conversely, RACK1 positively correlates with MAPK signaling in tumorigenesis (Sato et al, 2015; Song et al, 2024), but the impact of USP17LA on MAPK signaling in T cells remains unexplored. Given that USP17LA interacts with numerous proteins, it likely regulates additional substrates beyond RACK1. Among the 47 proteins associated with cadherin-binding and calmodulin-binding pathways, MYH9 was identified as a ubiquitination-regulated target analyzed by Uniprot. While MYH9 mutations are linked to hearing loss in humans with no significant immune implications (Li et al, 2016), uncharacterized ubiquitination of other proteins may play a role in T-cell modulation.

USP17LA and USP17LB exhibit high amino acid sequence homology, differing primarily at their N- and C-termini, suggesting that these structural domains may confer distinct regulatory functions in T cells. Consistent with this notion, although both USP17LA and USP17LB inhibited NFAT activity in HEK293T overexpression reporter assays, they appeared to have opposite effects on overall T-cell activation. This paradox may reflect the limitations of the artificial overexpression system, which lacks the full TCR signaling context and may obscure functional differences. USP17LA appears to regulate T-cell activation more broadly by stabilizing RACK1 and modulating MAPK and NF-κB pathways, while USP17LB may differ in subcellular localization, protein interactions, or expression dynamics, which could result in distinct or even compensatory functions under specific stimuli or cellular states. For instance, USP17LB may not associate with key regulators such as RACK1 to the same extent, potentially accounting for the

divergent signaling outcomes. These observations highlight the functional divergence within the USP17L family and underscore the need for further investigation into their context-dependent roles in T-cell biology.

Owing to the conserved nature of the USP17L family, we hypothesized that compensatory mechanisms might operate among its members. To address this, we assessed the expression of other USP17L genes in CD3⁺ T cells lacking either USP17LA or USP17LB. Under resting conditions, no significant changes were detected, implying limited baseline compensation (Fig. EV5C,D). However, upon TCR engagement using anti-CD3 and anti-CD28 antibodies, stimulus-dependent changes emerged. In *Usp17la*-deficient T cells, *Usp17ld* and *Usp17le* were markedly upregulated, suggestive of a compensatory attempt to restore regulatory balance in the absence of USP17LA. In contrast, *Usp17lb* deficiency resulted in elevated *Usp17la* and reduced *Usp17le* expression, indicating an alternative compensatory axis (Fig. EV5C,D). These findings underscore a dynamic, context-dependent regulatory interplay among USP17L family members and highlight the complexity of DUB-mediated modulation of T-cell activation. *Usp17la/Usp17lb* double KO mice may serve as valuable models to investigate potential compensatory mechanisms between these two homologs, which warrants further investigation in future studies. Given that ubiquitination is orchestrated by the coordinated actions of E3 ligases and DUBs, future studies are warranted to define the specific E3 partners of USP17LA and their contributions to TCR signaling fidelity.

Immune checkpoint inhibitors (ICIs) targeting PD-1 (CD279), PD-L1 (CD274), or CTLA-4 (CD152) have revolutionized cancer immunotherapy, leading to significant survival benefits in multiple cancer types (Wallis et al, 2019). However, many patients do not respond to ICI monotherapy (Wallis et al, 2019), highlighting the need to identify additional regulatory mechanisms controlling T-cell function in anti-tumor immunity. Recent advances suggest that targeting T-cell homeostasis in combination with ICIs may enhance therapeutic efficacy (Chen and Mellman, 2017; Hashimoto et al, 2023; Luo et al, 2023). Given that *Usp17la* knockout mice exhibit enhanced anti-tumor immunity, we hypothesize that targeting USP17LA in combination with ICIs could yield improved therapeutic outcomes. Further studies are required to validate this hypothesis and explore the underlying molecular mechanisms.

In summary, our study demonstrates that *Usp17la* deficiency enhances T-cell activation by modulating TCR signaling. We further show that USP17LA depletion suppresses tumor growth, underscoring its potential as a target for cancer immunotherapy. Future studies should explore whether USP17LA inhibition can

synergize with ICIs and further elucidate the downstream molecular pathways governing T-cell activation and tumor immunity.

# Methods

## Reagent and tools table

| Reagent/resource | Reference or source | Identifier or catalog number |
|---|---|---|
| **Experimental models** | | |
| C57BL/6J mice | Shanghai Model Organisms Center | NM-KO-200804 |
| OT-I mice | Prof. Bing Du from East China Normal University | N/A |
| MC38 | American Type Culture Collection | N/A |
| EL4 | American Type Culture Collection | N/A |
| HEK293T | American Type Culture Collection | N/A |
| **Recombinant DNA** | | |
| pLV-EF1a-N-Flag (EV) | This study | N/A |
| pLV-EF1a-N-Flag-USP17LA | This study | N/A |
| pLV-EF1a-N-Flag-USP17LB | This study | N/A |
| pLV-EF1a-N-Flag-RACK1 | This study | N/A |
| pLV-EF1-H1-NC | This study | N/A |
| pLV-EF1-H1-shUsp17la | This study | N/A |
| LentiCRISPR sg-Renilla | This study | N/A |
| LentiCRISPR sg-Usp17la | This study | N/A |
| pGL4.30-luc2P-NFAT-RE-Hygro | This study | N/A |
| pLV-luc2P-NFAT-RE | This study | N/A |
| pLV-puro-C-HA (EV) | This study | N/A |
| pLV-puro-C-HA-K48 | This study | N/A |
| pLV-puro-N-Myc | This study | N/A |
| pLV-puro-N-Myc-USP17LA | This study | N/A |
| **Antibodies** | | |
| APC/Fire™ 750 anti-mouse CD3ε | BioLegend | 100362 |
| FITC anti-mouse CD4 | BioLegend | 100509 |
| PE anti-mouse CD8α | BioLegend | 100707 |
| Brilliant Violet 785™ anti-mouse/human CD45R/B220 | BioLegend | 103246 |
| APC anti-mouse NK-1.1 | BioLegend | 108709 |
| APC anti-mouse/human CD11b | BioLegend | 101212 |
| Brilliant Violet 421™ anti-mouse F4/80 | BioLegend | 123137 |
| APC anti-mouse/human CD44 | BioLegend | 103011 |
| PE anti-mouse CD25 | BioLegend | 102007 |
| PE anti-mouse CD62L | BioLegend | 104408 |
| FITC anti-mouse IL-2 | BioLegend | 503806 |
| PE anti-mouse IL-4 | BioLegend | 504103 |
| PE anti-mouse IL-17A | BioLegend | 506903 |
| PE anti-mouse/human FOXP3 | BioLegend | 320207 |
| PE/Cyanine7 anti-human/mouse Granzyme B | BioLegend | 372213 |
| PE anti-mouse IFN-γ | BioLegend | 505807 |
| Brilliant Violet 650™ anti-mouse TNF-α | BioLegend | 506333 |

| Reagent/resource | Reference or source | Identifier or catalog number |
|---|---|---|
| Ultra-LEAF™ Purified anti-mouse CD3ε | BioLegend | 100340 |
| Ultra-LEAF™ Purified anti-mouse CD28 | BioLegend | 102116 |
| Ultra-LEAF™ Purified anti-mouse IL-4 | BioLegend | 504122 |
| Ultra-LEAF™ Purified anti-mouse IFN-γ | BioLegend | 505834 |
| Anti-β-actin | Sigma-Aldrich | A2228 |
| Anti-HSP90 | Proteintech | 13171-1-AP |
| Anti-Vinculin | Proteintech | 66305 |
| Anti-Flag | Sigma-Aldrich | F1804 |
| Anti-HA | Sigma-Aldrich | H9658 |
| Anti-c-Myc | Sigma-Aldrich | M4439 |
| Anti-p-ERK | Santa Cruz | sc-7383 |
| Anti-p-JNK1/2/3 | ABclonal | AP0276 |
| Anti-p-P38 | ABclonal | AP4771 |
| Anti-phospho-NF-κB P65 | Cell Signaling Technology | 3033 T |
| Anti-p-IKKα/β (S176/180) | Cell Signaling Technology | 2697 T |
| Anti-ERK | ABclonal | A4782 |
| Anti-JNK1/2/3 | ABclonal | A4867 |
| Anti-P38 | ABclonal | A4771 |
| Anti-P65 | Cell Signaling Technology | 8242T |
| Anti-IKKβ | Cell Signaling Technology | 8943T |

| Oligonucleotides and other sequence-based reagents | Forward | Reverse |
|---|---|---|
| Usp17la-RT | GAGGTCTTTGGAGACATGGTG | CCAACTCAGACTGTGCTTTCC |
| Usp17lb-RT | GTGGTTGCTCTCTCCTTCC | CTCTCCCAACTCAGACTGTG |
| Usp17lc-RT | GAGGTCTTTGGAGACATGGTG | TCAGACTGGGCTTGTCATTG |
| Usp17ld-RT | TGGTGGTTTCTCTTTCCTTCC | TCAGACTGGGCTTGTCATTG |
| Usp17le-RT | CAAGTTCTTTGAAGAGGTCTTTGG | AGACTGTGCTTTCCATTGGTAG |
| RACK1-RT | GTCCCGAGACAAGACCATAAA | GGACACAAGACACCCATTCT |
| ACTB-RT | ACCTTCTACAATGAGCTGCG | CTGGATGGCTACGTACATGG |
| sg-Usp17la | CACCGTGGAGGAGCTAACTGTCAA | AAACTTGACAGTTAGCTCCTCCAC |
| sh-Usp17la | AAAA GCTGTAAGTTGTGTGCTATGG TTGGATCCAA CCATAGCACACAACTTACAGC | AAAA GCTGTAAGTTG TGTGCTATGG TTGGA TCCAA CCATAGCACAC AACTTACAGC |
| sh-Usp17lb | AAAA GGTGGAGGTCTCAGATCAAGT TTGGATCCAA ACTTGATCTGAGACCTCCACC | AAAA GGTGGAGGT CTCAGATCAAGT TTGGATCCAA ACTTGATCTGAGA CCTCCACC |
| Usp17la-genotyping (F1/F2/R1) | TCTCTCTACTTTGGTGGTCG (F1) GCTGACTGACTCTCTGATTG (F2) | CTCTATGCTGCT CAGATTCC (R1) |
| Usp17lb-genotyping (F3/F4/R2) | TGGTCATGAGTTG AAGCCGT (F3) GAACTTGATCGACT CAGTGG (F4) | AGAAAGGGCAGTCAC AAAGC (R2) |
| Usp17la-genotyping (q-F1/q-R1) | CTTCTATGTGCAGCA GGCCA | TGTTTTCGCAGGGCTC TCCTAA |
| Usp17lb-genotyping (q-F2/q-R2) | GTGGTTGCTCTCTC CTTCC | CTCTCCCAACTCAG ACTGTG |
| **Chemicals, enzymes, and other reagents** | | |
| True-Nuclear™ Transcription Factor Buffer Set | BioLegend | 424401 |

| Reagent/resource | Reference or source | Identifier or catalog number |
|---|---|---|
| DNase I | Sigma-Aldrich | 10104159001 |
| Collagenase D | Sigma-Aldrich | 11088858001 |
| Percoll | Cytiva | 17089109-1 |
| EDTA | Sangon Biotech | B540625-0500 |
| Nonidet (R) P-40 | Sangon Biotech | A500109-0500 |
| 2 x Taq Plus Master Mix II | Vazyme | P213-03 |
| PrimeSTAR Max Premix (2X) | TAKARA | R045Q |
| MojoSort™ Mouse CD3 T Cell Isolation Kit | BioLegend | 480031 |
| MojoSort™ Mouse CD4 Naive T Cell Isolation Kit | BioLegend | 480040 |
| MojoSort™ Mouse CD8 Naive T Cell Isolation Kit | BioLegend | 480044 |
| DMEM | Sangon Biotech | E600003-0500 |
| RPMI 1640 | Sangon Biotech | E600028-0500 |
| Fetal bovine serum | Lonsera | S711-001S |
| Penicillin–streptomycin solution | Sangon Biotech | E607011-0500 |
| Nonessential amino acids | Beyotime | C0332 |
| Sodium pyruvate | BasalMedia | 113-24-6 |
| PMA | Sigma-Aldrich | P8139 |
| Dextran Sulfate Sodium Salt | MP Biomedicals | 0216011010 |
| Ionomycin | Sigma-Aldrich | 407952 |
| Monensin | Peprotech | 2237803 |
| Immunostaining Fix Solution | Beyotime | P0098 |
| Permeabilization Solution with Saponin | Beyotime | P0095 |
| CountBright Absolute Counting Beads | Thermo Fisher | C36950 |
| CFDA SE Cell Proliferation and Tracking Kit | Yeasen | 40714ES76 |
| ConA | Sigma-Aldrich | 11028-71-0 |
| LDH Cytotoxicity Assay Kit | Beyotime | C0017 |
| BCA Protein Assay Kit | Epizyme | ZJ102 |
| HiScript III RT SuperMix for qPCR Kit | Vazyme | R323-01 |
| Anti-Flag M2 magnetic beads | Sigma-Aldrich | M8823 |
| 3× FLAG peptide | Sigma-Aldrich | F4799 |
| Protein Stains O | Sangon Biotech | C510027 |
| Protease Inhibitor Cocktail | Sigma-Aldrich | P2714 |
| Nitrocellulose membranes | Cytiva | 10600002 |
| *Escherichia coli* Stellar cells | Clontech | 636763 |
| SYBR qPCR Master Mix | Vazyme | Q712 |
| Recombinant Human TGF-β1 | BioLegend | 580702 |
| Human IL-2 | Novoprotein | C013 |
| Mouse IL-2 | Peprotech | 212-12 |
| Mouse IL-4 | Peprotech | 214-14 |
| Mouse IL-12 | Peprotech | 210-12 |
| Recombinant Mouse IL-6 (carrier-free) | BioLegend | 575702 |
| **Software** | | |
| FlowJo v10.6 | TreeStar | |

| Reagent/resource | Reference or source | Identifier or catalog number |
|---|---|---|
| GraphPad Prism 8 | GraphPad Software | |
| **Other** | | |
| Celesta cell analyzer | BD Biosciences | |
| Orbitrap Exploris 480 | Thermo Fisher | |
| Pierce™ ECL Western Blotting Substrate | Thermo Fisher | |
| ChemiScope 6100 | Tanon | |
| NanoDrop 2000 spectrophotometer | Thermo Fisher | |

## Mice

All mice used in this study were on a C57BL/6J background, with age and gender matched across groups. Male and female mice were used interchangeably unless otherwise noted. The *Usp17la*-knockout (*Usp17la*$^{-/-}$) and *Usp17lb*-knockout (*Usp17lb*$^{-/-}$) mice were generated by the Shanghai Model Organisms Center, Inc. using a CRISPR/Cas9−mediated gene targeting strategy. The genotype of mice was confirmed by PCR of mouse tail DNA and RT-PCR using cDNA from CD3$^+$ T cells isolated from the spleen. OT−I mice were originally provided by Prof. Bing Du (East China Normal University). *Usp17la*$^{-/-}$ OT−I mice were obtained by crossing *Usp17la*$^{-/-}$ mice with OT−I transgenic mice. To minimize bias and experimental variability, mice were randomly grouped and re-labeled with randomized identification numbers. All mice were bred and maintained under specific pathogen-free conditions, with ad libitum access to standard laboratory chow, a 12 h light/dark cycle, controlled temperature (22 ± 1 °C), and humidity (55% ± 5%), in accordance with the guidelines approved by the Animal Studies Committee of Children's Hospital of Fudan University. All in vivo experiments were conducted following protocols approved by the Institutional Animal Care and Use Committee of Fudan University (Protocol No. 2020IBSJS-013).

## Primary cells and cell lines

Primary immune cell suspensions from thymus, spleen, inguinal lymph nodes, and blood were prepared as described (Zhang et al, 2025). In brief, cells from the thymus, spleen, and lymph nodes were harvested by mechanical grinding, while peripheral blood was collected via the submandibular vein. After erythrocyte lysis from the spleen and peripheral blood, the resulting cell suspensions were used for peripheral immune cell analysis. For in vitro T-cell activation, Th polarization and TCR signal analysis, primary CD3$^+$ T cells, naive CD4$^+$ T cells or naive CD8$^+$ T cells were isolated from the spleens of mice using negative selection kits, according to the manufacturer's protocol. To isolate tumor-infiltrating lymphocytes (TILs), MC38 tumors were harvested from mice and subjected to enzymatic digestion with DNase I and collagenase D for 30 min at 37 °C with gentle rotation. The resulting cell suspension was then separated by density gradient centrifugation using a 20%/80% Percoll solution. The lymphocyte-enriched fraction was collected from the interphase and subsequently resuspended in complete culture medium for further analysis.

The mouse lymphoma (EL4) cells and human embryonic kidney (HEK293T) cells were sourced from the American Type Culture Collection. Both EL4 and HEK293T cells were cultured in DMEM

medium supplemented with 10% fetal bovine serum and 1% penicillin–streptomycin. Primary cell lines were cultured in RPMI 1640 medium, supplemented with 10% FBS, nonessential amino acids, sodium pyruvate, and 55 μM β-mercaptoethanol. All cells were maintained in a humidified incubator with 5% $CO_2$ at 37 °C. Cell authentication was performed via short tandem repeat (STR) profiling.

## Flow cytometry

Flow cytometry was performed using a Celesta cell analyzer (BD Biosciences), and data were analyzed with FlowJo software. Immune cells were stained at 4 °C with FACS buffer (2% BSA in 1× PBS) and analyzed using the following surface markers: CD3ε, CD4, CD8, B220, NK1.1, CD11b, F4/80, CD44, CD25, and CD62L. For intracellular cytokine analysis, cells were stimulated with PMA and ionomycin in the presence of Monensin for 4–5 h. Intracellular markers included IL-2, IL-4, IL-17A, Granzyme B, IFN-γ, and TNF-α. Intracellular cytokine staining was conducted at 4 °C using Immunostaining Fix Solution and Permeabilization Solution with Saponin. For transcription factor analysis, Foxp3 staining was performed at 4 °C using the True-Nuclear Transcription Factor Buffer Set. For cell quantification, CountBright Absolute Counting Beads were added to the samples prior to flow cytometric analysis.

## Th cell polarization

According to the previous description, naive CD4+ T cells were isolated from mouse spleens using a negative selection kit and activated with 2 μg/mL of plate-bound anti-CD3 and 1 μg/mL of anti-CD28. To induce polarization into specific subsets, the following conditions were applied: Th1 cells were polarized with 10 μg/mL anti-IL-4 and 10 ng/mL IL-12; Th2 cells were induced with 10 μg/mL anti-IFN-γ and 20 ng/mL IL-4; Th17 cells were differentiated using 20 ng/mL IL-6, 5 ng/mL TGF-β, along with anti-IFN-γ and anti-IL-4; and iTreg cells were differentiated with 1 ng/mL TGF-β, 4 ng/mL IL-2, and anti-IFN-γ and anti-IL-4. After 72 h of stimulation, Monensin was added for the last 4 h to block protein transport, and cells were then collected for flow cytometry analysis.

## Carboxyfluorescein succinimidyl ester (CFSE) cell proliferation

CFSE dilution assays were performed using the CFDA SE Cell Proliferation and Tracking Kit. The cell concentration was adjusted to $1–5 × 10^6$ cells/mL, and cells were stained with a 2× CFDA SE working solution at 37 °C for 15 min. The staining reaction was then terminated by adding 5–10 times the volume of medium supplemented with 10% FBS, following the manufacturer's protocol. CFSE-stained cells were cultured with ConA for 48 or 72 h, and cell division was subsequently analyzed by flow cytometry.

## B16F10 lung metastasis model

A total of $2 × 10^5$ B16/F10 melanoma cells, suspended in 100 μL PBS, were i.v. injected into 6–8-week-old WT and $Usp17la^{-/-}$ female mice. Fourteen days post-injection, the mice were anesthetized and their lungs were carefully excised for examination of metastatic nodules and photographic documentation.

## Subcutaneous syngeneic MC38 colon adenocarcinoma model

A total of $1 × 10^6$ MC38 cells suspended in 100 μL PBS were injected subcutaneously into 6–8-week-old WT and $Usp17la^{-/-}$ mice. Tumor volume was measured every 3 days starting 8 days post-injection. Tumor volume was calculated using the formula: ½ × length × width², where length and width represent the longest and shortest tumor diameters, respectively. After 25 days, the tumors were excised and weighed. CD4+ and CD8+ tumor-infiltrating T cells were isolated for subsequent flow cytometry analysis.

## In vitro cytotoxicity assay

Total splenocytes from WT OT−I and $Usp17la^{-/-}$ OT−I mice were isolated and cultured in vitro with $10^{-7}$ M $OVA_{257-264}$ peptide for 24 h, following lysis of red blood cells. The cells were then maintained in IL-2 for an additional 4 days to generate effector cells. On day 5, $2 × 10^4$ B16/F10 cells were seeded in a 96-well plate and pulsed with $10^{-7}$ M $OVA_{257-264}$ peptide for 1 h to serve as target cells. The activated WT and $Usp17la^{-/-}$ OT−I cells were subsequently collected and incubated with peptide-pulsed B16/F10 cells at varying effector-to-target ratios for 4 h. Cell death was quantified using the LDH Cytotoxicity Assay Kit at an optical density of 490 nm (OD490) and calculated according to the manufacturer's instructions.

## DSS-induced colitis model

To induce acute colitis, 8−10-week-old male WT and $Usp17la^{-/-}$ mice on a C57BL/6 background were given 1.5% dextran sulfate sodium (DSS; molecular weight 36,000–50,000) in their drinking water for 7 consecutive days, followed by 3 days of regular water. Body weight was monitored daily starting from the first day of DSS treatment.

## Luciferase reporter assay

For HEK293T cell-based luciferase reporter assay, HEK293T cells were co-transfected with luciferase reporter plasmids and either (empty vector) EV, USP17LA, USP17LB or DUSP2 expression constructs. After 24 h, cells were stimulated with PMA and ionomycin for 4 h, lysed at room temperature for 10 min, and centrifuged to collect the supernatant. Luciferase activity was measured using a commercial luciferase assay system according to the manufacturer's instructions.

For the EL4 cell-based luciferase reporter assay, EL4 cells were transduced with lentivirus encoding an NFAT-luciferase reporter cassette, and stable cell lines were established. These cell lines were subsequently transduced with lentivirus expressing $Usp17la$ shRNA and/or RACK1 overexpression constructs. After 3 days, cells were stimulated with PMA and ionomycin for 3 or 6 h, followed by lysis and measurement of luciferase activity.

## Immunoprecipitation and co-immunoprecipitation/mass spectrometry (Co-IP/MS)

Transfected HEK293T cells were harvested and lysed in NP-40 lysis buffer (25 mM Tris-HCl, pH 7.5, 150 mM NaCl, 1% NP-40/Triton X-100, 1 mM EDTA, 1 mM EGTA, 2.5 mM sodium pyrophosphate,

1 mM β-glycerophosphate, 1 mM Na$_3$VO$_4$, 1 mM PMSF) supplemented with protease inhibitors. Cell lysates were centrifuged at 12,000× $g$ for 10 min and incubated with anti-Flag M2 magnetic beads for 2–4 h at 4°C. After extensive washing, the anti-Flag-beads −bound proteins were eluted using 3× FLAG peptide. The eluted proteins were then subjected to immunoblotting or silver staining prior to semiquantitative mass spectrometry analysis by the Functional Proteomics Analysis Platform, Institutes of Biomedical Sciences, Fudan University.

## Ubiquitination assay

HEK293T cells were transfected with HA-Ub (WT or K48) and the indicated constructs. After 36 h, the cells were treated with 20 µM MG132 for 4 h, then lysed in denaturing buffer [1% (wt/vol) SDS, 50 mM Tris-HCl, pH 7.5, 0.5 mM EDTA, and 1 mM DTT]. The lysates were heated at 100 °C for 5 min, sonicated, and then diluted 10-fold with NP-40 lysis buffer. After centrifugation, the lysates were incubated with anti-Flag M2 beads for 2 h at 4 °C. Following extensive washing, the bead-bound proteins were eluted with 3× FLAG peptide and resolved by SDS-PAGE for subsequent immunoblotting.

## Immunoblotting

Immunoblotting was performed as previously described (Zhang et al, 2025). Briefly, cells or tissues were lysed in RIPA lysis buffer (25 mM Tris-HCl, pH 7.5, 150 mM NaCl, 1% NP-40/Triton X-100, 1 mM EDTA, 1% sodium deoxycholate, 0.1% SDS), supplemented with protease inhibitors, 1 mM PMSF, and 1 mM Na$_3$VO$_4$. Protein concentrations were determined using the BCA Protein Assay Kit, resolved by sodium dodecyl sulfate−polyacrylamide gel electrophoresis (SDS-PAGE), and transferred to Nitrocellulose membranes. After blocking with 5% skim milk, the membranes were sequentially incubated with primary and secondary antibodies. The primary antibodies used included: anti-β-actin; anti-HSP90; anti-Vinculin; anti-Flag; anti-HA; anti-c-Myc; anti-p-ERK;anti-p-JNK1/2/3; anti-p-P38; anti-phospho-NF-κB P65 (Ser536), p-IKKα/β (S176/180); ERK; JNK1/2/3; P38; P65; IKKβ. Bands were visualized using Pierce™ ECL Western Blotting Substrate with a ChemiScope 6100 (Tanon, Shanghai, China).

## Plasmid constructs

Human gene *RACK1* was amplified by PCR and subcloned into the pLV-EF1a-IRES-puro vector with an N- terminal Flag tag. Mouse gene *Usp17la* and *Usp17lb* were conducted by amplifying cDNA from mouse tissues mix and cloning it into the pLV-EF1a-IRES-puro vector with an N-terminal Myc tag. K48 Ub was constructed by mutating six other lysine residues (K6, K11, K27, K29, K33, and K63) to arginine and subsequently cloned into the pcDNA6-N-HA vector. To knockout endogenous *Usp17la*, lentiviral sgRNA vectors were generated by inserting the sgRNA sequences into the lentiCRISPR v2 vector. Lentiviral shRNA vectors targeting endogenous *Usp17la* were generated by cloning the shRNA sequences into the pLV-EF1a-H1 vector. shRNA sequences were designed using Thermo Fisher's BLOCK-iT RNAi Designer tool. All plasmids were transformed into *Escherichia coli* Stellar cells, and plasmid DNA was extracted. All plasmid constructs were validated by DNA sequencing.

## RNA isolation and quantitative real-time PCR (qRT-PCR)

Total RNA was extracted from cultured cells using TRIzol reagent in accordance with the manufacturer's instructions. The quality and concentration of the RNA were assessed using a NanoDrop 2000 spectrophotometer. Subsequently, RNA was transcribed into first-strand complementary DNA (cDNA) as needed, utilizing the HiScript III RT SuperMix for qPCR Kit. Quantitative real-time PCR (RT-qPCR) was conducted using SYBR qPCR Master Mix on a QuantStudio 1 system (Applied Biosystems) following the manufacturer's protocol. The expression levels of the target genes were normalized to β-actin using the $2^{-\Delta\Delta Ct}$ method.

## Bioinformatics analyses

The RNA-seq data from EL4 cells stimulated with PMA and Ionomycin were analyzed using differential expression data reported in our previous study (Zhang et al, 2019). Tissue expression profile data for the *Usp17l* family were obtained from a previous study ((Söllner et al, 2017); Data Ref: (Söllner et al, 2017)). For the analysis of USP17LA IP/MS data, two sets of data were integrated: one comprising proteins with a USP17LA-to-vector ratio greater than two, and the other derived from the USP17LA-only group. Tissue expression profiles were evaluated using the STRING database, and Gene Ontology (GO) molecular function enrichment analysis was performed using the DAVID database. Proteins involved in the calmodulin binding and cadherin-binding pathways, as identified through DAVID, were further analyzed and visualized using a specialized bioinformatic tool (http://www.bioinformatics.com.cn).

## Statistical analysis

Data are presented as the mean ± standard deviation (s.d.), as reflected by the error bars in the graphs. Sample sizes were determined to ensure sufficient statistical power for detecting potential differences. Statistical significance was determined using GraphPad Prism 9.0 software. Group differences were assessed with an unpaired two-tailed Student's $t$ test for comparisons between two groups and with ordinary one-way or two-way analysis of variance (ANOVA) for comparisons involving multiple groups. A $P$ value of >0.05 was deemed not significant. Some samples were excluded from the analysis based on pre-established criteria, such as poor viability, failed stimulation. These exclusions were made prior to unblinding and were necessary to ensure data quality.

# Data availability

The mass spectrometry proteomics data have been deposited to the ProteomeXchange Consortium via the PRIDE partner repository with the dataset identifier PXD065847.

The source data of this paper are collected in the following database record: biostudies:S-SCDT-10_1038-S44319-025-00584-5.

# Peer review information

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

## Acknowledgements

We thank Prof. Bing Du (East China Normal University) for generously providing OT−I TCR transgenic mice. This work was supported by the National Natural Science Foundation of China (grant numbers 32370926 and 82071780) and the Science and Technology Commission of Shanghai Municipality (grant number 21JC1400900).

## Author contributions

**Huiling Zhang**: Conceptualization; Resources; Data curation; Formal analysis; Validation; Investigation; Visualization; Writing—original draft; Project administration; Writing—review and editing. **Zhihan Guo**: Resources; Data curation; Formal analysis; Validation; Investigation; Visualization; Writing—original draft. **Gaigai Wei**: Validation; Investigation. **Jingjing Yi**: Validation; Investigation. **Zixi Wang**: Validation; Investigation. **Yuqi Zhang**: Validation; Investigation. **Haiping Zhao**: Validation; Investigation. **Tingrong Ren**: Validation; Investigation. **Yihan Wang**: Validation; Investigation. **Jiating Kuang**: Validation; Investigation. **Zhaoying Sheng**: Validation; Investigation. **Duanwu Zhang**: Conceptualization; Resources; Supervision; Funding acquisition; Visualization; Project administration; Writing—review and editing.

Source data underlying figure panels in this paper may have individual authorship assigned. Where available, figure panel/source data authorship is listed in the following database record: biostudies:S-SCDT-10_1038-S44319-025-00584-5.

## Disclosure and competing interests statement

The authors declare no competing interests.

# Expanded View Figures

**Figure EV1.　Transcriptomic analysis of activated EL4 cells, sequence homology alignment of the USP17L family, and cDNA-based genotyping of *Usp17la* and *Usp17lb* knockout mice.** ▶

(A) GO pathway analysis of upregulated expressed genes in PMA plus ionomycin vs untreated EL4 cells by RNA-seq were conducted using the David database (https://david.ncifcrf.gov/). (B) KEGG pathway analysis of upregulated expressed genes in PMA plus ionomycin vs untreated EL4 cells by RNA-seq were conducted using the David database (https://david.ncifcrf.gov/). (C) Alignment was performed using the Clustal Omega program and reformatted with the MView tool. cov, percent coverage; pid, percent identity with respect to the first sequence (USP17LA). Identities normalized by aligned length. Colors indicate identical sequence. (D, E) Schematic diagram of cDNA-based genotyping for *Usp17la* and *Usp17lb* knockout mice. CD3$^+$ T cells were sorted from mouse spleens, followed by RNA extraction, reverse transcription, and genotyping using cDNA. (F, G) Representative RT-PCR results showing cDNA-based genotyping of *Usp17la$^{+/+}$* vs. *Usp17la$^{-/-}$* and *Usp17llb$^{+/+}$* vs. *Usp17lb$^{-/-}$* transcripts using RNA from CD3$^+$ T cells sorted from mouse spleens. (H) Genotypic distribution of 522 offspring from *Usp17la$^{+/-}$* × *Usp17la$^{+/-}$* crosses was compared with the expected Mendelian ratio (1:2:1) using a χ$^2$ goodness-of-fit test. No significant deviation was observed ($P > 0.05$). (I) Body weight of 8-week-old *Usp17la$^{+/+}$*, *Usp17la$^{+/-}$*, and *Usp17la$^{-/-}$* mice ($n = 5$ per genotype). (J) Gross appearance of 8-week-old *Usp17la$^{+/+}$* and *Usp17la$^{-/-}$* mice. Data are representative of two independent experiments (I). Data are shown as mean ± s.d. Statistical analyses were performed using one-way ANOVA with Sidak's multiple comparisons test (I). NS, not significant.

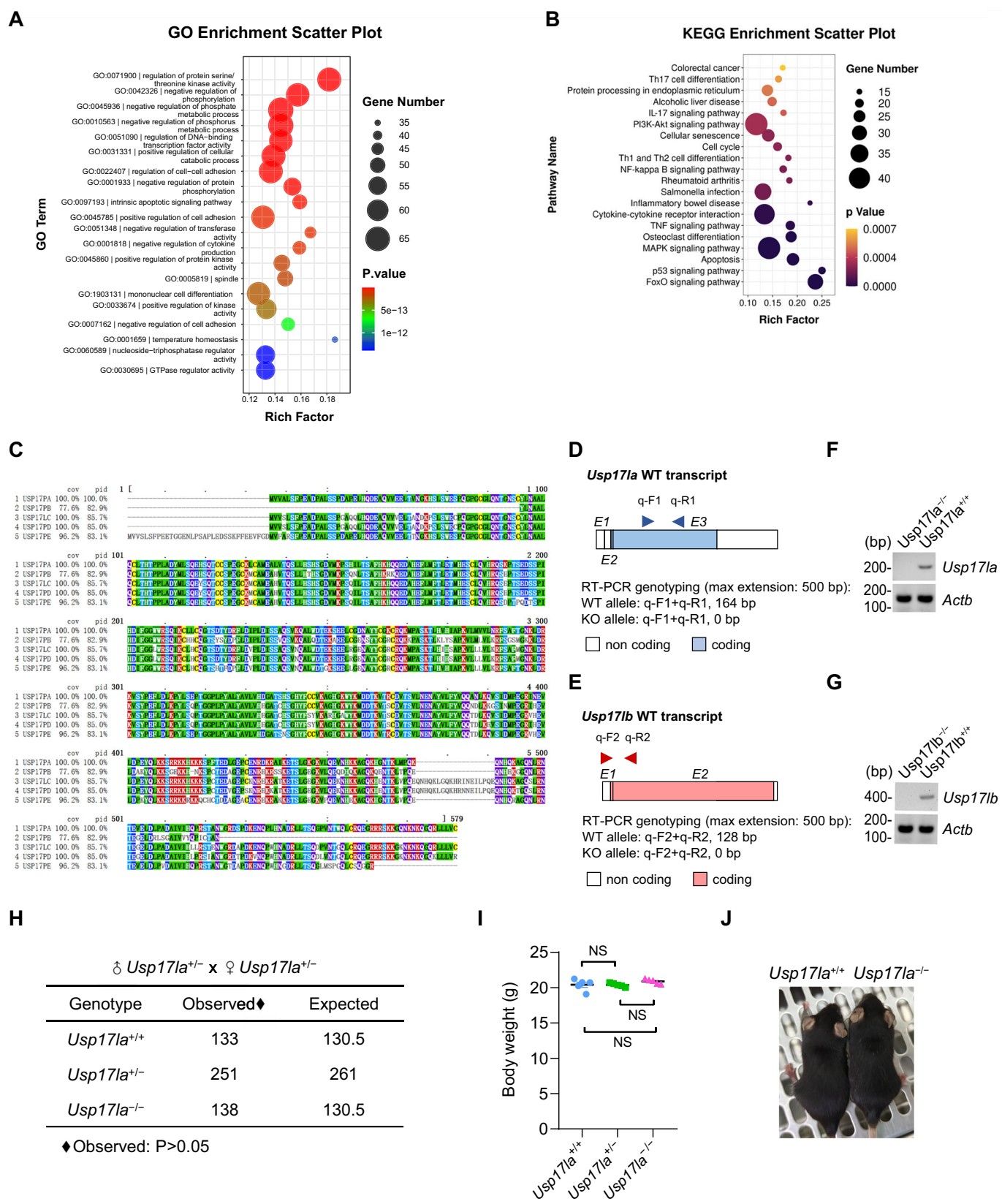

**A** GO Enrichment Scatter Plot

**B** KEGG Enrichment Scatter Plot

**C**

**D** *Usp17la* WT transcript

RT-PCR genotyping (max extension: 500 bp):
WT allele: q-F1+q-R1, 164 bp
KO allele: q-F1+q-R1, 0 bp

☐ non coding  ☐ coding

**E** *Usp17lb* WT transcript

RT-PCR genotyping (max extension: 500 bp):
WT allele: q-F2+q-R2, 128 bp
KO allele: q-F2+q-R2, 0 bp

☐ non coding  ☐ coding

**F**

**G**

**H**

♂ *Usp17la*⁺ᐟ⁻ x ♀ *Usp17la*⁺ᐟ⁻

| Genotype | Observed♦ | Expected |
|---|---|---|
| *Usp17la*⁺ᐟ⁺ | 133 | 130.5 |
| *Usp17la*⁺ᐟ⁻ | 251 | 261 |
| *Usp17la*⁻ᐟ⁻ | 138 | 130.5 |

♦Observed: P>0.05

**I**

**J**

*Usp17la*⁺ᐟ⁺  *Usp17la*⁻ᐟ⁻

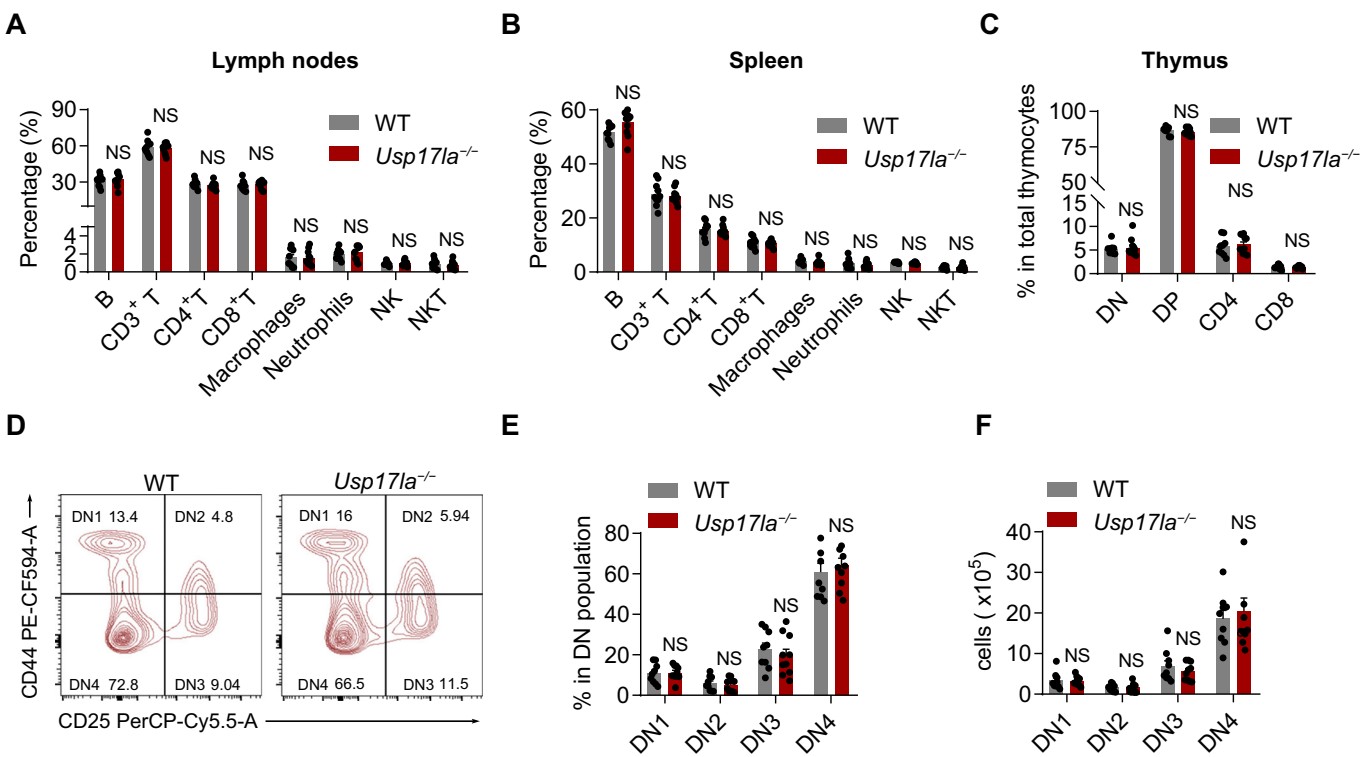

**Figure EV2. The percentage of immune cells in lymphoid organs and DN subpopulation in *Usp17la⁻/⁻* mice.**

(A, B) The percentage of B, CD3⁺ T, CD4⁺ T, CD8⁺ T, macrophages, neutrophils, NK and NKT cells in lymph nodes and spleen of WT and *Usp17la⁻/⁻* mice (*n* = 10–11 per genotype). (C) The percentage of DN, DP, CD4 SP, and CD8 SP subpopulations in thymus of WT and *Usp17la⁻/⁻* mice (*n* = 10–11 per genotype). (D–F) Flow cytometry analysis the frequency and number of DN1, DN2, DN3, and DN4 cells in DN population of WT and *Usp17la⁻/⁻* mice (*n* = 10–11 per genotype). Data are combination of two independent experiments. Data are shown as mean ± s.d. Statistical analyses were performed using two-way ANOVA followed by Sidak's multiple comparisons test (A–C, E, F). NS, not significant.

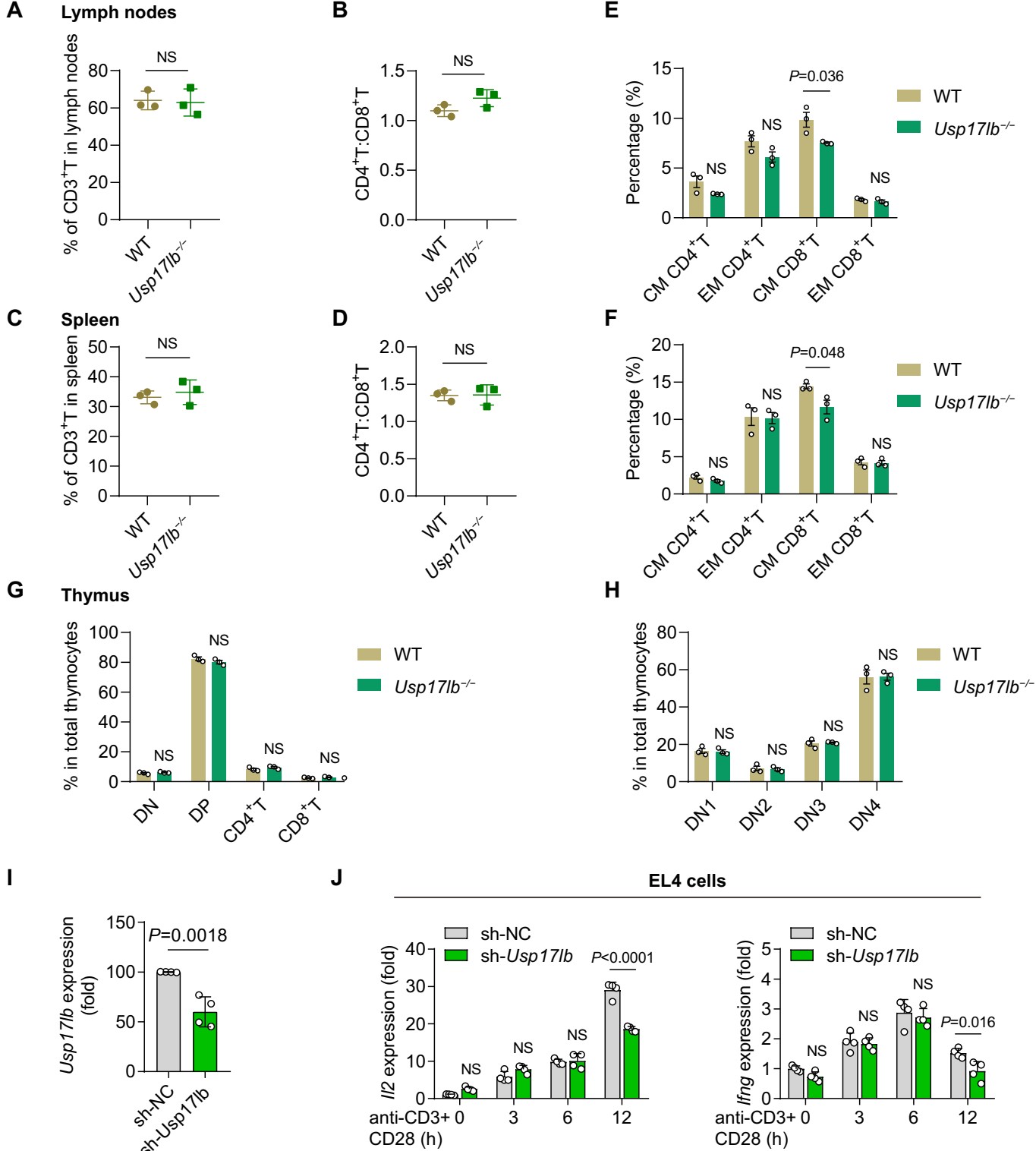

◄

**Figure EV3.  Immune profiling of *Usp17lb*<sup>−/−</sup> mice and functional activation of EL4 cells following *Usp17lb* loss.**

(A, B) Percentages of CD3$^+$ T cells in total lymphocytes and the ratio of CD4$^+$ T cells to CD8$^+$ T cells in lymph nodes of WT and *Usp17lb*$^{-/-}$ mice ($n = 3$ per genotype). (C, D) Percentages of CD3$^+$ T cells in total lymphocytes and the ratio of CD4$^+$ T cells to CD8$^+$ T cells in spleen of WT and *Usp17lb*$^{-/-}$ mice ($n = 3$ per genotype). (E, F) Frequencies of CM and EM subpopulations in CD4$^+$ T cells or CD8$^+$ T cells in lymph nodes and spleen of WT and *Usp17lb*$^{-/-}$ mice ($n = 3$ per genotype). (G, H) Flow cytometry analysis the percentages of CD4$^-$CD8$^-$ (DN), CD4$^+$CD8$^+$ (DP), CD4$^+$CD8$^-$ (SP4), and CD4$^-$CD8$^+$ (SP8) subpopulations in thymus, and CD44$^+$CD25$^-$ (DN1), CD44$^+$CD25$^+$ (DN2), CD44$^-$CD25$^+$ (DN3), and CD44$^-$CD25$^-$ (DN4) subpopulations among DN thymocytes of WT and *Usp17lb*$^{-/-}$ mice ($n = 3$ per genotype). (I) RT-PCR analysis of *Usp17lb* knockdown efficiency in EL4 cells following infection with sh-NC or sh-*Usp17lb* lentivirus ($n = 4$ per group). (J) RT-PCR analysis of the expression of *Il-2* and *Ifng* in EL4 cells infected with sh-NC or sh-*Usp17lb* lentivirus following anti-CD3 plus CD28 stimulation for the indicated times ($n = 4$ per group). Data are representative of three (**A–H**) or two independent experiments (**I, J**). Data are shown as mean ± s.d. Statistical analyses were performed using unpaired, two-tailed Student's *t*-test (**A–D, I**) and two-way ANOVA followed by Sidak's multiple comparisons test (**E–H, J**). NS not significant.

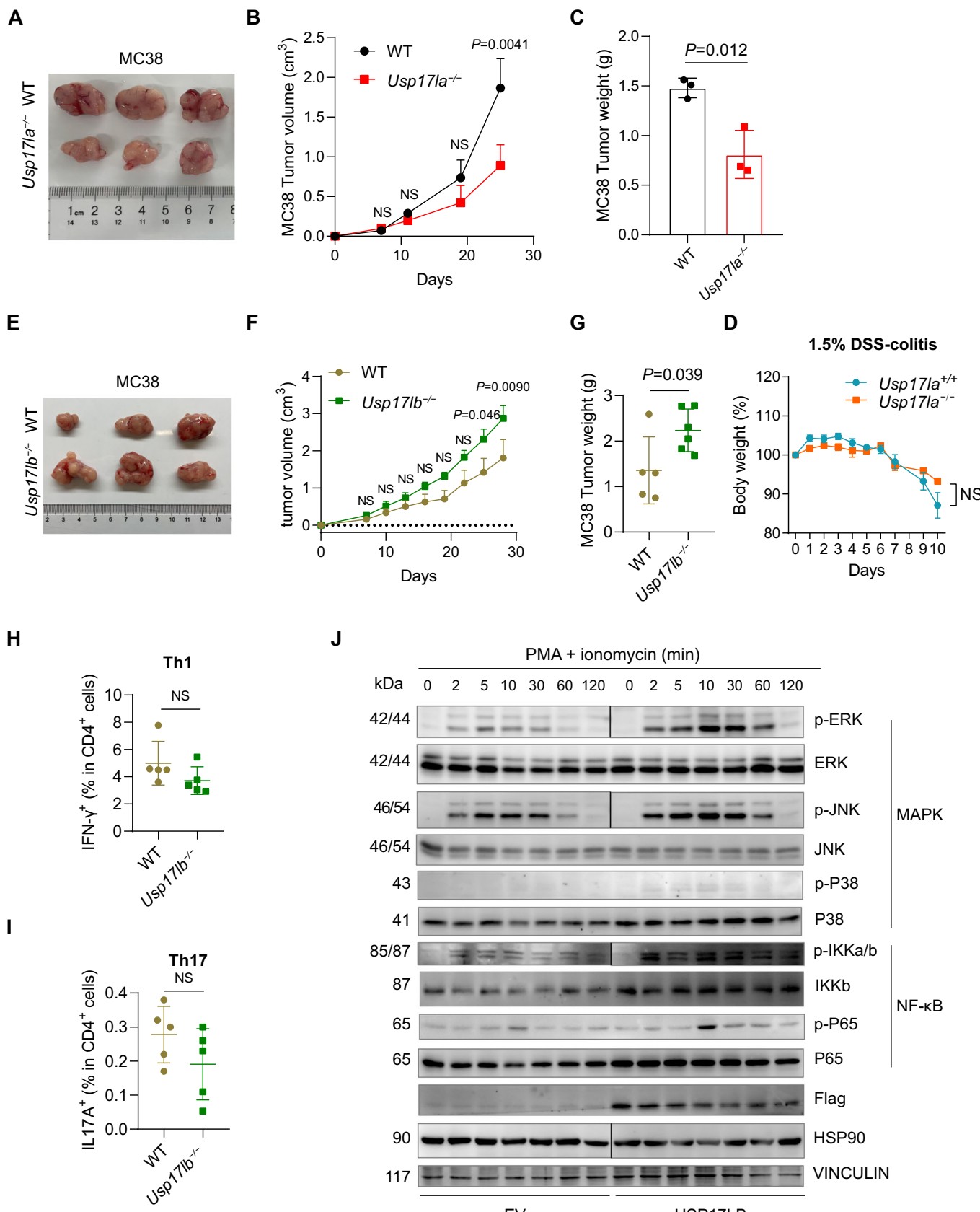

◀  **Figure EV4.  *Usp17la* deficiency suppresses tumor growth, whereas *Usp17lb* deficiency promotes it and its overexpression augments TCR signaling.**

(A–C) Tumor growth was assessed in terms of the tumor volume (**A**), tumor size (**B**), and tumor weight (**C**) in WT and *Usp17la*$^{-/-}$ mice after MC38 colon cancer cells inoculation ($n = 3$ per genotype). (**D**) Body weight changes in DSS-induced colitis. Mice were administered 1.5% DSS in drinking water for 7 days, followed by a 3-day recovery period with DSS-free water. Body weight was recorded over a total of 10 days. ($n = 4$ per genotype). (**E–G**) Tumor growth was assessed in terms of the tumor volume (**E**), tumor size (**F**), and tumor weight (**G**) in WT and *Usp17lb*$^{-/-}$ mice after MC38 colon cancer cells inoculation (for **E**, $n = 3$ per genotype; for (**F, G**), $n = 5$–6 per genotype). (**H, I**) Flow cytometry analysis of the expression of IFN-γ and IL-17A in CD4$^+$ T cells from spleen of WT and *Usp17lb*$^{-/-}$ mice ($n = 5$ per genotype).
(**J**) Immunoblot analysis of TCR signaling-associated proteins in PMA plus ionomycin-stimulated EL4 cells overexpressed with empty control (EV) or Flag-USP17LB. Data are representative of two independent experiments (**A–E, H–J**) or pooled from two independent experiments (**F, G**). Data are shown as mean ± s.d. Statistical analyses were performed using two-way ANOVA followed by Sidak's multiple comparisons test (**B, D, F**) and unpaired, two-tailed Student's *t* test (**C, G, H, I**). NS, not significant. Source data are available online for this figure

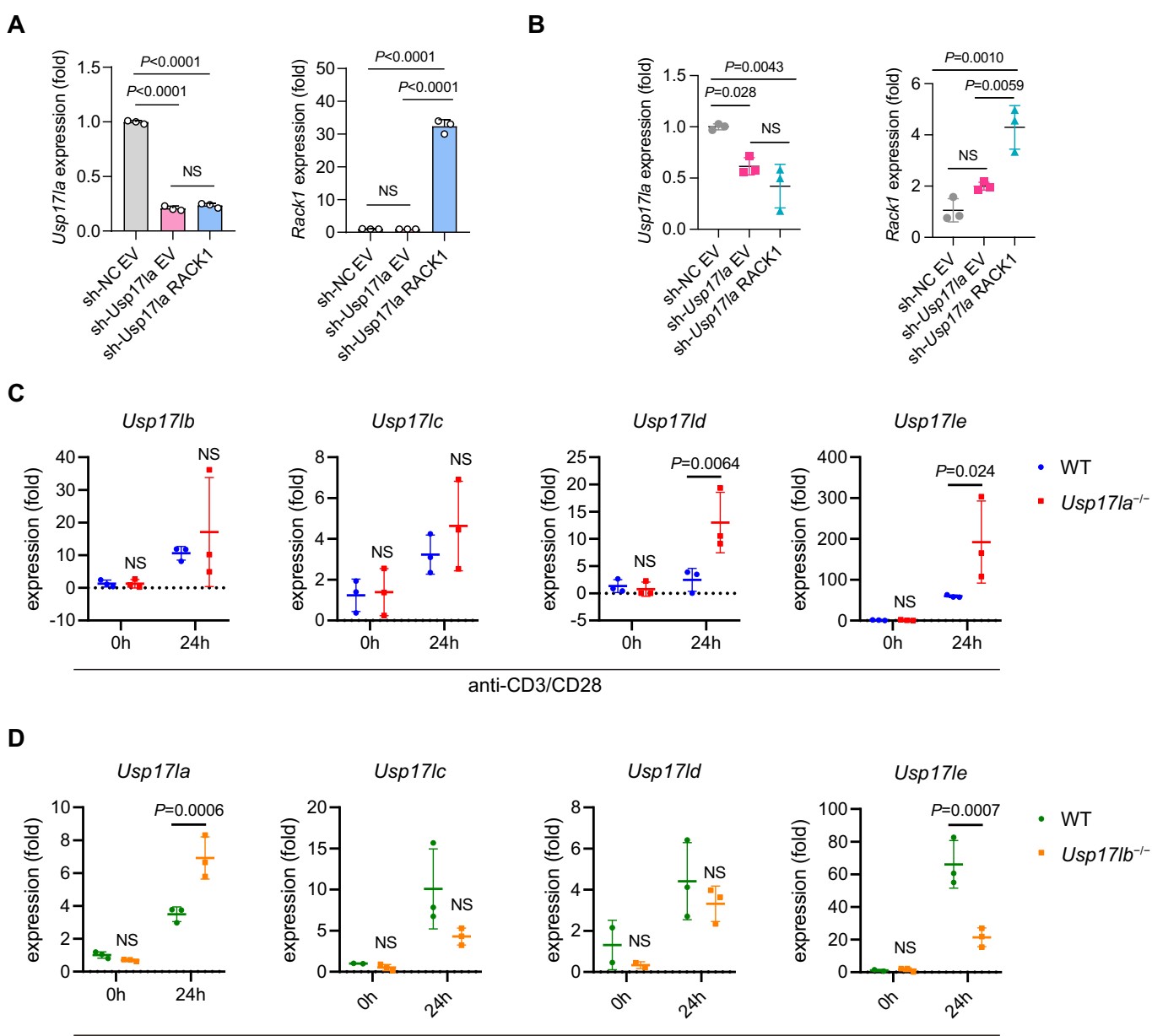

**Figure EV5. Compensatory effects induced by *Usp17la* and *Usp17lb* deficiency.**

(A) RT-PCR analysis of *Usp17la* and *Rack1* mRNA expression in EL4 cells following infection with sh-NC or sh-*Usp17la* in combination with OE-EV or OE-RACK1 lentiviruses (n = 3 per group). (B) RT-PCR analysis of *Usp17la* and *Rack1* mRNA expression in stably NFAT-luciferase-transduced EL4 cells following infection with sh-NC or sh-*Usp17la* in combination with OE-EV or OE-RACK1 lentiviruses (n = 3 per group). (C, D) RT-PCR analysis of other *Usp17l* family members in *Usp17la* or *Usp17lb*-deficient cells with or without anti-CD3 plus anti-CD28 stimulation for 24 h (n = 3 per group). Data are representative of two independent experiments (A–D). Data are shown as mean ± s.d. Statistical analyses were performed using one-way ANOVA with Dunnett's multiple comparisons test (A, B) and unpaired, two-tailed Student's *t* test (C, D). NS not significant.

