## [Peer Review File · EMBO Reports]

The deubiquitinase USP17LA negatively regulates T-cell activation and attenuates anti-tumor immunity

Huiling Zhang, Duanwu Zhang, Zhihan Guo, Gaigai Wei, Zixi Wang, Jingjing Yi, Yuqi Zhang, Haiping zhao, Tingrong Ren, Yihan Wang, Jiating Kuang, and Zhaoying Sheng

Corresponding author(s): Duanwu Zhang (duanwu@fudan.edu.cn)

Review Timeline:

Submission Date:	20th Feb 25
Editorial Decision:	14th Apr 25
Revision Received:	22nd Jun 25
Editorial Decision:	7th Aug 25
Revision Received:	18th Aug 25
Accepted:	5th Sep 25

Transaction Report:

Dear Prof. Zhang

Thank you for the submission of your research manuscript to our journal. I sincerely apologize for the delay in handling your manuscript, but we have now received the two referee reports copied below.

As you will see, the referees acknowledge that the findings are potentially interesting and that part of the conclusions are overall supported by the data presented but they also raise a number of concerns and have suggestions how to further strengthen the data that should be addressed. Referee 2 suggested to test a potential redundant role of Usp171a and Usp171b in double KO mice. While these experiments would without doubt give interesting further insight, they appear beyond the scope of the current study and are not required for publication here (as also indicated by referee 2 him/herself).

Given these constructive comments, we would like to invite you to revise your manuscript with the understanding that the referee concerns (as detailed above and in their reports) must be fully addressed and their suggestions taken on board. Please address all referee concerns in a complete point-by-point response. Acceptance of the manuscript will depend on a positive outcome of a second round of review. It is EMBO Reports policy to allow a single round of revision only and acceptance or rejection of the manuscript will therefore depend on the completeness of your responses included in the next, final version of the manuscript.

I am also happy to discuss the revision further via e-mail or a video call, if you wish.

We realize that it is difficult to revise to a specific deadline. In the interest of protecting the conceptual advance provided by the work, we recommend a revision within 3 months (July 14). Please discuss the revision progress ahead of this time with the editor if you require more time to complete the revisions.

=====

IMPORTANT NOTE:

We perform an initial quality control of all revised manuscripts before re-review. Your manuscript will FAIL this control and the handling will be delayed IN CASE the following APPLIES:

- 1) A data availability section providing access to data deposited in public databases is missing. If you have not deposited any data, please add a sentence to the data availability section that explains that.
- 2) Your manuscript contains statistics and error bars based on $n=2$. Please use scatter blots in these cases. No statistics should be calculated if $n=2$.

=====

- 1) a .docx formatted version of the manuscript text (including legends for main figures, EV figures and tables). Please make sure that the changes are highlighted to be clearly visible.
- 2) individual production quality figure files as .eps, .tif, .jpg (one file per figure). Please download our Figure Preparation Guidelines (figure preparation pdf) from our Author Guidelines pages <https://www.embopress.org/page/journal/14693178/authorguide> for more info on how to prepare your figures.
- 3) a .docx formatted letter INCLUDING the reviewers' reports and your detailed point-by-point responses to their comments. As part of the EMBO Press transparent editorial process, the point-by-point response is part of the Review Process File (RPF), which will be published alongside your paper.
- 4) a complete author checklist, which you can download from our author guidelines (<<https://www.embopress.org/page/journal/14693178/authorguide>>). Please insert information in the checklist that is also reflected in the manuscript. The completed author checklist will also be part of the RPF.
- 5) Please note that all corresponding authors are required to supply an ORCID ID for their name upon submission of a revised

manuscript (<<https://orcid.org/>>). Please find instructions on how to link your ORCID ID to your account in our manuscript tracking system in our Author guidelines (<<https://www.embopress.org/page/journal/14693178/authorguide#authorshipguidelines>>)

6) We replaced Supplementary Information with Expanded View (EV) Figures and Tables that are collapsible/expandable online. A maximum of 5 EV Figures can be typeset. EV Figures should be cited as 'Figure EV1, Figure EV2' etc... in the text and their respective legends should be included in the main text after the legends of regular figures.

7) Please note that a Data Availability section at the end of Materials and Methods is now mandatory. In case you have no data that requires deposition in a public database, this should still be stated as "This study includes no data deposited in external repositories."

See also < <https://www.embopress.org/page/journal/14693178/authorguide#dataavailability>>. Please note that the Data Availability Section is restricted to new primary data that are part of this study.

Additional information on source data and instruction on how to label the files are available <<https://www.embopress.org/page/journal/14693178/authorguide#sourcedata>>

10) Figure legends and data quantification:
The following points must be specified in each figure legend:

- the name of the statistical test used to generate error bars and P values,
 - the EXACT p-values,
 - the number (n) of independent experiments (please specify technical or biological replicates) underlying each data point,
 - the nature of the bars and error bars (s.d., s.e.m.)
- If the data are obtained from n {less than or equal to} 5, show the individual data points in addition to the SD or SEM.
- If the data are obtained from n {less than or equal to} 2, use scatter blots showing the individual data points.

See also the guidelines for figure legend preparation:
<https://www.embopress.org/page/journal/14693178/authorguide#figureformat>

11) Our journal encourages inclusion of *data citations in the reference list* to directly cite datasets that were re-used and obtained from public databases. Data citations in the article text are distinct from normal bibliographical citations and should directly link to the database records from which the data can be accessed. In the main text, data citations are formatted as follows: "Data ref: Smith et al, 2001" or "Data ref: NCBI Sequence Read Archive PRJNA342805, 2017". In the Reference list, data citations must be labeled with "[DATASET]". A data reference must provide the database name, accession number/identifiers and a resolvable link to the landing page from which the data can be accessed at the end of the reference. Further instructions are available at <<https://www.embopress.org/page/journal/14693178/authorguide#referencesformat>>.

12) All Materials and Methods need to be described in the main text using our 'Structured Methods' format. According to this format, the Methods section includes a Reagents and Tools Table (listing key reagents, experimental models, software and relevant equipment and including their sources and relevant identifiers) followed by a Methods and Protocols section describing the methods, ideally using a step-by-step protocol format. The aim is to facilitate adoption of the methodologies across labs.

Please download and fill our Reagents and Tools Table template (.docx), which you can find in our author guidelines: <https://www.embopress.org/page/journal/14693178/authorguide#structuredmethods>.

An example of a Method paper with Structured Methods can be found here: <https://www.embopress.org/doi/10.15252/msb.20178071>.

13) As part of the EMBO publication's Transparent Editorial Process, EMBO Reports publishes online a Review Process File to accompany accepted manuscripts. This File will be published in conjunction with your paper and will include the referee reports, your point-by-point response and all pertinent correspondence relating to the manuscript.

Yours sincerely,

=====

Referee #1:

The manuscript USP17LA Restrains T cell Activation and Anti-Tumor Immunity by Targeting the MAPK/NFKB/NFAT axis by Zhang et al. reports that knocking out the deubiquitinase enzyme USP17LA in mice results in increased activation of T cells through reduced expression of the NFAT inhibitor, RACK1. The investigation of USP17LA in T cells is novel and the findings are potentially significant to the understanding of T cell activation, which is relevant for immune modulation in broad contexts, including anti-tumor immunity. The studies should be of broad interest to the molecular biology community because the results explore molecular mechanisms regulating NFAT activity and activation of cytokine expression. The results strongly support that perturbing USP17LA enhances CD4 and CD8 T cell cytokine production when stimulated in vitro (Figure 3). In addition, the overexpression and knockdown studies in cell lines also support that USP17LA inhibits NFAT, and loss of USP17LA can be rescued by overexpressing the NFAT inhibitor RACK1 (Figures 6-7). While these findings are intriguing, other aspects of the studies are not of sufficient quality to justify the conclusions regarding T cell activity in vivo and anti-tumor immunity. It should not be too difficult or time consuming to complete additional experiments needed to support these conclusions. Although unacceptable in its present form, the study is sufficiently promising to encourage resubmission in the future.

Major Concerns:

1. The knockout mice generated by Crispr have not been adequately validated. KO was only shown by PCR from tail biopsies. The authors need to do something to verify the KO mice- perhaps QPCR to look for transcripts in T cells, as well as specifying the primer locations for the genotyping PCR that would explain why the KO band is larger than the WT band. It is essential that the KO mice be verified beyond tail PCR.
2. The authors state that the USP17LA KO had no effect on T cell development, differentiation, or homeostasis. However, the data used to rule out these other aspects of T cell biology is not sufficiently powered. In Figure 2, only 4-5 mice per genotype are used. In Figure 4, only 3-4 mice are used in the tumor models. In some places, the authors claim differences between groups when there is not statistically significant difference (Figure 4F, S5C, S5E), or claim no differences when there are statistically significant changes (S4G and S4H). Additional mice should be added to these in vivo studies to ensure the conclusions are robust, and the language in the text needs to be accurate when describing results. In addition, the in vitro differentiation studies (Figure 5) have a high degree of variability (some experiments showed little or no differentiation), and these also appear underpowered. Additional mice should be added to the in vivo studies, especially the tumor experiments to ensure the conclusions about T cell activation are robustly supported. Information about the sex of the mice should be included.

3. The title is misleading- only NFAT activation is impacted by USP17LA. MAPK and NFkB activity were not increased, and these should be removed from the title.
4. Although USP17LA regulates NFAT and RACK1 rescues cytokine production in USP17LA KO, but whether RACK1 rescues NFAT activity was not determined. Defining whether RACK1 overexpression rescues NFAT activity in USP17LA KO cells would strengthen the main conclusion that USP17LA suppresses activation of T cells through stabilizing expression of RACK1, an NFAT inhibitor.

Minor concerns:

1. The finding that USP17LA and USP17LB both inhibit NFAT yet have opposing roles in T cell activation warrants more discussion.
2. Weight loss after DSS treatment should not be used to make conclusions about T cell differentiation in vivo.
3. There needs to be more information in the Methods about: DSS colitis, bioinformatics analyses used (Figure 1C, 7A-C), primary T cell isolation for in vitro activation, shUSP17LA constructs.
4. If T cells were isolated from spleens of USP17LA KO mice, which have fewer Tregs, were Tregs removed from the cultures or did fewer Tregs contribute to the increases in cytokine production? This should be discussed.
5. The labels in the figure legends do not match the figures in several cases (Figures 1, 2, 4)
6. Lines 266-272 should move to the discussion.
7. The authors state that they did not have antibodies that could distinguish between USP17LA and USP17LB, which is why they could not verify the KO by immunoblot, so they should remove the statement from the Methods that says they verified the KO by immunoblot.
8. Results with USP17LB don't add much to the story and are also underpowered- even the most compelling piece of data (S5A-C) has only 2 replicates. Some of these data could be shored up to draw a contrast between the roles of USP17LA and USP17LB or much of this could be omitted to shorten the manuscript.

Referee #2:

The authors investigated the role of the USP17L family in T cell regulation, based on that several members of these deubiquitinases were up-regulated after PMA/ionomycin in the EL-4 mouse T lymphoblast cell line. They proceeded to make knock outs of two of the USP17L family members and found that one of them caused higher T cell responses and improved anti-cancer CD8 T cell effects, but not autoimmune CD4 T-cell driven effects in a model of IBD. Mechanistically, they suggest that USP17LA regulate T cell responses via regulating NFAT activity and suggest a mechanism for this effect. The authors have thus found a novel regulatory mechanism for T cell activation. The figures are clear and informative.

The authors selected to study USP17LA and USP17LB based on expression in thymus and level of induction in EL-4 cells after PMA/I stimulation. Since at USP17LC, USP17LD and USP17LE also were upregulated in EL-4 cells, it could be interesting to at least partially address the potential for redundancy between the different USP17L family members.

An obvious question is of course if Usp17la/Usp17lb double mutants show additional phenotypes not seen in the respective single mutant parents. It could be that the lack of Usp17lb phenotype is because Usp17la is dominant, but that the Usp17lb phenotype becomes apparent in the double knock out. This would indicate redundancies between A and B. This might however not be absolutely needed to resolve in the scope of this paper.

Another question is whether the other USP17L variants are stronger induced in stimulated T cells that are knocked out for either USP17LA or USP17LB, or both. A stronger induction in the knock out backgrounds could indicate redundancy and an attempt to compensate for the loss of the major USP17L variant in T cells.

Minor comments:

row 110-111 : The term "high homology" is wrong. Homology is qualitative (common evolutionary ancestry? yes/no). You probably mean "high sequence similarity".

Response to Reviewers:

We sincerely thank the reviewers for their thorough evaluation of our manuscript titled "USP17LA Restrains T-Cell Activation and Anti-Tumor Immunity by Targeting the MAPK/NF- κ B/NFAT Axis". We appreciate the constructive feedback and have carefully revised the manuscript accordingly. Below we provide a point-by-point response to each comment. All revisions are highlighted in yellow in the revised manuscript. Additionally, we have adjusted certain formatting elements in accordance with the journal's requirements.

Referee #1:**Major Concerns:**

1. The knockout mice generated by Crispr have not been adequately validated. KO was only shown by PCR from tail biopsies. The authors need to do something to verify the KO mice- perhaps QPCR to look for transcripts in T cells, as well as specifying the primer locations for the genotyping PCR that would explain why the KO band is larger than the WT band. It is essential that the KO mice be verified beyond tail PCR.

Response: We fully agree with you that further validation of the *Usp17la* and *Usp17lb* KO mice is essential beyond genotyping PCR from tail biopsies.

In the revised manuscript, we have now included RT-PCR data from purified CD3⁺ T cells, demonstrating the absence of *Usp17la* and *Usp17lb* transcripts in KO mice (Fig EV1D–G). In wild-type mice, specific PCR bands can be amplified. In contrast, due to the deletion of the target gene, *Usp17la*- and *Usp17lb*-deficient mice failed to produce the corresponding bands, thereby allowing clear distinction between wild-type and knockout genotypes. These results confirm the successful deletion of target gene transcripts in T cells and further verify the KO model at the mRNA level.

Besides, to clarify the PCR-based genotyping strategy, we have also revised schematic diagrams of the CRISPR/Cas9-mediated KO design and the positions of the genotyping primers for both *Usp17la* and *Usp17lb* (Fig 1F, G). In each case, two guide RNAs were designed to excise the entire coding sequence (CDS) of the target gene. Genotyping

was performed using three primers: one common reverse primer and two gene-specific forward primers. PCR conditions were optimized to generate amplicons under 700 bp.

- For *Usp17la*:
WT allele: F2 + R1 → 138 bp
KO allele: F1 + R1 → 264 bp
- For *Usp17lb*:
WT allele: F4 + R2 → 356 bp
KO allele: F3 + R2 → 572 bp

This strategy explains the increased size of the KO bands, which varies according to the gene locus and the primer pair used. The clear difference in amplicon size enables precise distinction between WT and KO alleles.

Together, these additions confirm the successful deletion of *Usp17la* and *Usp17lb* at both the genomic and transcript levels, thereby reinforcing the genetic integrity and reliability of our knockout mouse models. This enhanced validation provides a robust foundation for the downstream functional analyses and strengthens the overall rigor of our study. We sincerely thank the reviewer for this valuable suggestion, which has significantly improved the quality of our data.

2. The authors state that the USP17LA KO had no effect on T cell development, differentiation, or homeostasis. However, the data used to rule out these other aspects of T cell biology is not sufficiently powered. In Figure 2, only 4-5 mice per genotype are used. In Figure 4, only 3-4 mice are used in the tumor models. In some places, the authors claim differences between groups when there is not statistically significant difference (Figure 4F, S5C, S5E), or claim no differences when there are statistically significant changes (S4G and S4H). Additional mice should be added to these in vivo studies to ensure the conclusions are robust, and the language in the text needs to be accurate when describing results. In addition, the in vitro differentiation studies (Figure 5) have a high degree of variability (some experiments showed little or no differentiation), and these also appear underpowered. Additional mice should be added to the in vivo studies, especially the tumor experiments to ensure the conclusions about T cell activation are robustly supported. Information about the sex of the mice should be included.

Response: We appreciate these important suggestions. To enhance statistical power and ensure the robustness of our conclusions, we have increased the number of mice analyzed in key experiments, including peripheral T cell homeostasis (Fig 2A–J, P, Q, EV2A, B), thymocyte development (Fig 2K–O, EV2C–F), Th differentiation (Fig 5D), and tumor models (Figs 4E–G, EV4A–C, F, G). Additionally, we combined data from two independent EL4 cell activation assays following *Usp17la* or *Usp17lb* knockdown (Fig 3C, D and EV3I, J). All corresponding quantifications and statistical analyses have been updated accordingly. Additionally, we carefully re-evaluated all statistical descriptions and corrected the figure legends and main text where appropriate to ensure consistency and accuracy. These revisions have been clearly highlighted in the revised manuscript.

In Fig 2A–J, EV2A, B, integrated results from two independent experiments (approximately 10–11 mice per genotype) consistently demonstrated that *Usp17la* deficiency does not affect the number or proportion of major immune cell subsets—including B cells, T cells, NK cells, NKT cells, macrophages, neutrophils—in lymphoid organs. Similarly, the activation state of T cells under homeostatic conditions remained unaffected in Fig. 2P and Q (7 mice per genotype). Moreover, combined data from two independent experiments (approximately 10–11 mice per genotype) in Fig 2K–O, EV2C–F further supported that thymocyte development is not altered by *Usp17la* deficiency.

For the Th differentiation assay (Fig 5C, D), additional biological replicates were included (approximately 5–6 mice per genotype). To improve data accuracy, several outliers and samples that failed to respond to stimulation were excluded from the analysis. The revised data confirm that *Usp17la* deficiency does not impact Th cell differentiation.

In the MC38 tumor model, data from two independent experiments (approximately six mice per genotype) were combined for Fig. 4E–G, with additional analyses of tumor growth and weight presented in Fig. EV4A–C. These results consistently support the conclusion that *Usp17la* deficiency enhances anti-tumor immunity. Similarly, for the MC38 model using *Usp17lb*-deficient mice, two independent experiments were combined (approximately five to six mice per genotype), and the findings in Fig. EV4F–G consistently demonstrated that *Usp17lb* deficiency promotes tumor growth.

For the EL4 cell activation assay, we combined results from two independent experiments. The data showed that *Usp17la* knockdown promotes T cell activation, whereas *Usp17lb* knockdown reduces activation status.

Lastly, the sex of the mice used is now clearly stated in the revised Methods section.

We believe that these revisions have significantly improved the rigor and clarity of our study. We sincerely appreciate your thoughtful and constructive feedback. We hope that our responses and the additional data provided have fully addressed your concerns and strengthened the overall conclusions of the manuscript.

3. The title is misleading- only NFAT activation is impacted by USP17LA. MAPK and NFkB activity were not increased, and these should be removed from the title.

Response: We sincerely thank you for this insightful comment and the opportunity to clarify our findings. However, we respectfully believe that the interpretation that "only NFAT activation is impacted" may arise from a misunderstanding. Our data clearly demonstrated that USP17LA deficiency enhances two major TCR downstream signaling pathways—MAPK and NF- κ B—as evidenced by increased phosphorylation of ERK and P65, respectively. In parallel, USP17LA overexpression suppressed NFAT transcriptional activity, further supporting its regulatory role across multiple signaling axes.

Specifically, in Fig 6A–D, we showed that phosphorylation of ERK and P65 was significantly increased in both primary T cells and EL4 cells following USP17LA deletion or knockdown, under anti-CD3/CD28 or PMA/ionomycin stimulation. In Fig 6E–F, the luciferase reporter assay in HEK293T cells demonstrated reduced NFAT activity upon USP17LA overexpression. Taken together, these findings support our conclusion that USP17LA negatively regulates T cell activation via coordinated modulation of the MAPK, NF- κ B, and NFAT pathways.

In our study, IP/MS analysis revealed that USP17LA interacts with a broad range of proteins. Among these, we identified the NFAT–RACK1 regulatory axis as one potential mechanism underlying the observed phenotypes. While our data support this pathway as a key contributor, we acknowledge that additional mechanisms—particularly those modulating the MAPK and NF- κ B pathways—may also be involved.

However, a comprehensive exploration of these alternative mechanisms is beyond the scope of the current study and will be the focus of future research. We hope this explanation addresses your concern, and we sincerely appreciate your thoughtful feedback.

4. Although USP17LA regulates NFAT and RACK1 rescues cytokine production in USP17LA KO, but whether RACK1 rescues NFAT activity was not determined. Defining whether RACK1 overexpression rescues NFAT activity in USP17LA KO cells would strengthen the main conclusion that USP17LA suppresses activation of T cells through stabilizing expression of RACK1, an NFAT inhibitor.

Response: We appreciate this valuable suggestion. In response, we performed additional experiments to assess whether RACK1 overexpression can inhibit NFAT activity in USP17LA-deficient cells. Specifically, we established EL4 cell lines stably expressing an NFAT-luciferase reporter construct. In these cells, *Usp17la* was knocked down using shRNA, followed by reconstitution of RACK1 through lentiviral transduction. The cells were then stimulated with anti-CD3 and anti-CD28 antibodies to activate TCR signaling, and NFAT activity was measured via luciferase assay. Our results showed that *Usp17la* knockdown significantly enhanced NFAT reporter activity, indicating heightened downstream TCR signaling. Notably, reintroduction of RACK1 markedly attenuated this increase, suggesting that RACK1 can partially rescue the elevated NFAT activation induced by USP17LA deficiency (Fig. 7H). These findings support our proposed model in which USP17LA negatively regulates NFAT signaling by stabilizing RACK1.

We have added the relevant data and corresponding descriptions to the revised Results section of the manuscript, which are highlighted for your convenience. We hope that these additional experiments and mechanistic insights adequately address your comment and further reinforce the conclusions of our study.

Minor concerns:

1. The finding that USP17LA and USP17LB both inhibit NFAT yet have opposing roles in T cell activation warrants more discussion.

Response: We thank you for this insightful comment. As suggested, we have expanded the discussion in the revised manuscript to address the functional divergence between USP17LA and USP17LB. Although both proteins inhibited NFAT activity in HEK293T overexpression reporter assays, they exhibited opposing effects on overall T cell activation. This apparent paradox likely reflects the limitations of the artificial assay system, which does not recapitulate the full complexity of TCR signaling. Our data suggest that USP17LA exerts broader inhibitory effects on T cell activation by stabilizing RACK1 and suppressing downstream MAPK and NF- κ B pathways. In contrast, USP17LB may differ in its subcellular localization, interaction networks, or expression dynamics, resulting in distinct regulatory roles. For instance, USP17LB may not engage key signaling regulators such as RACK1 to the same extent, or may act in a compensatory or context-dependent manner under specific cellular conditions. We have now included this expanded discussion in the revised Discussion section.

2. Weight loss after DSS treatment should not be used to make conclusions about T cell differentiation in vivo.

Response: We agree and have removed the interpretation linking DSS-induced weight loss to T cell differentiation. We now present these data as evidence of colitis severity only in revised Fig EV4D, without over-interpreting the mechanism.

3. There needs to be more information in the Methods about: DSS colitis, bioinformatics analyses used (Figure 1C, 7A-C), primary T cell isolation for in vitro activation, shUSP17LA constructs.

Response: We appreciate your constructive comment. In response, we have substantially revised and expanded the Methods section to provide comprehensive experimental details for the following components:

- DSS-induced colitis model: We now describe the DSS concentration (1.5%), treatment duration (7 days followed by 3 days of normal water), mouse strain and age, and daily body weight monitoring, used to evaluate colitis susceptibility in WT and *Usp17la*^{-/-} mice.
- Bioinformatics analyses: We have specified the RNA-seq dataset source (Zhang

et al., 2019), the differential expression thresholds applied, integration strategies for USP17LA IP/MS datasets, and the use of STRING and DAVID databases for protein interaction and GO enrichment analyses. Calmodulin and cadherin-binding proteins were further analyzed using <http://www.bioinformatics.com.cn>.

- Primary T cell isolation: Naïve CD4⁺, naïve CD8⁺, or CD3⁺ T cells were isolated from mouse spleens using commercial negative selection kits. These primary T cells were subsequently used for in vitro T cell activation, Th cell polarization, and TCR signaling assays. In particular, detailed protocols for Th cell polarization—including stimulation conditions, cytokine cocktails, and antibody treatments—have been fully described, with relevant reagent sources and catalog numbers provided.
- shUSP17LA construct: We have included the vector backbone (pLV-EF1a-H1), shRNA design tool (BLOCK-iT, Thermo Fisher), and cloning strategy used to generate the lentiviral shUSP17LA constructs.

We thank the reviewer for the valuable suggestion to improve our manuscript. We believe these additions enhance methodological transparency and reproducibility, in line with your recommendation.

4. If T cells were isolated from spleens of USP17LA KO mice, which have fewer Tregs, were Tregs removed from the cultures or did fewer Tregs contribute to the increases in cytokine production? This should be discussed.

Response: We thank you for raising this important point. To clarify, we re-evaluated the proportion of Treg cells in the spleens of USP17LA-deficient mice and found no significant differences compared with wild-type controls across two independent experiments. In CD4⁺ T cell activation assays, Treg cells were not specifically depleted and may have been present in the cultures. However, considering that USP17LA deficiency does not affect Treg frequency in the spleen, nor does it impair iTreg differentiation in vitro or alter Treg distribution under either homeostatic or tumor-bearing conditions (see Fig 2J, 4F, 5A–D), we believe that the presence of Tregs in these cultures is comparable between genotypes and thus unlikely to confound the cytokine production results. Nevertheless, the conclusion that *Usp17la* deficiency has no impact on CD4⁺ T cell activation should be considered with caution, given the

potential influence of Treg cells.

Furthermore, in naive CD8⁺ T cell activation assays, Treg cells were effectively removed through negative selection during cell isolation. Therefore, the enhanced cytokine responses observed in these assays were independent of any potential influence from Tregs.

We have incorporated this point into the Discussion section as suggested. Thank you for your valuable comment, and we hope our response addresses your concern.

5. The labels in the figure legends do not match the figures in several cases (Figures 1, 2, 4).

Response: We sincerely apologize for the oversight. In response to your comment, we have thoroughly reviewed all figure legends and corresponding figure panels, especially Figures 1, 2, and 4, to ensure consistency and accuracy. All mismatches have been corrected, and the updated figures and legends are included in the revised manuscript.

6. Lines 266-272 should move to the discussion.

Response: We appreciate your suggestion. As recommended, we have moved the content originally presented in lines 266–272 to the Discussion section in the revised manuscript, where it is now more appropriately placed within the context of interpreting our findings.

7. The authors state that they did not have antibodies that could distinguish between USP17LA and USP17LB, which is why they could not verify the KO by immunoblot, so they should remove the statement from the Methods that says they verified the KO by immunoblot.

Response: We thank you for pointing this out. As we indeed lacked specific antibodies to distinguish USP17LA or USP17LB from other members of the USP17L family, we were unable to verify the knockout at the protein level. Accordingly, we have removed the statement regarding immunoblot verification from the Methods section. Instead, we

now clarify that *Usp17la* knockout was confirmed by PCR of mouse tail DNA and RT-PCR using cDNA from CD3⁺ T cells isolated from the spleen. The relevant text under “Mice” section has been revised accordingly. We apologize for the oversight and sincerely appreciate your careful review.

8. Results with USP17LB don't add much to the story and are also underpowered—even the most compelling piece of data (S5A-C) has only 2 replicates. Some of these data could be shored up to draw a contrast between the roles of USP17LA and USP17LB or much of this could be omitted to shorten the manuscript.

Response: We thank you for the constructive comment regarding the USP17LB data. In the revised manuscript, we have clarified that the data presented in Fig EV3I and J are based on two independent experiments that were combined and analyzed together to increase statistical power and reliability. This has been explicitly stated in the revised figure legend.

In addition, in light of Reviewer #2's suggestion and to more clearly highlight the functional specificity of USP17LA, we believe it is valuable to show that other closely related family members, such as USP17LB, do not share the same regulatory effects on T cell activation. This distinction not only underscores the non-redundant role of USP17LA but also suggests the complexity and divergence of function within the USP17L family. Including USP17LB as a comparison thus enhances the overall clarity and specificity of our conclusions.

We hope that this explanation clarifies our rationale for retaining the USP17LB data, and we sincerely appreciate your thoughtful feedback which helped us improve the presentation and interpretation of these results.

Referee #2:

An obvious question is of course if *Usp17la/Usp17lb* double mutants show additional phenotypes not seen in the respective single mutant parents. It could be that the lack of *Usp17lb* phenotype is because *Usp17la* is dominant, but that the *Usp17lb* phenotype becomes apparent in the double knock out. This would indicate redundancies between A and B. This might however not be absolutely

needed to resolve in the scope of this paper.

Response: We thank you for this constructive suggestion. We fully agree that analysis of *Usp17la/Usp17lb* double knockout (DKO) mice would provide deeper insights into potential functional redundancy between these two family members, particularly given the possibility that USP17LA may exert a dominant role that masks the phenotype of USP17LB deficiency.

However, due to time and resource constraints, we have not yet generated DKO mice in the current study. We acknowledge this as a limitation and are actively planning to breed *Usp17la/Usp17lb* DKO mice in future work to further elucidate their potential cooperative or compensatory roles in regulating T cell activation. As also noted by the editor, while such experiments would indeed add depth to our understanding, they go beyond the scope of the current study, which focuses on defining the individual contributions of USP17LA and USP17LB.

We have now clearly acknowledged this limitation in the revised Discussion section. Once again, we sincerely thank you for your insightful comment, and we apologize for not being able to address this question experimentally at this stage.

Another question is whether the other USP17L variants are stronger induced in stimulated T cells that are knocked out for either USP17LA or USP17LB, or both. A stronger induction in the knock out backgrounds could indicate redundancy and an attempt to compensate for the loss of the major USP17L variant in T cells.

Response: We appreciate your important suggestion. To explore potential compensatory regulation and functional redundancy among USP17L family members, we sorted CD3⁺ T cells from WT, *Usp17la*-KO, and *Usp17lb*-KO mice and stimulated them with anti-CD3 and anti-CD28 antibodies for 0 or 24 hours. We then analyzed the mRNA expression levels of *Usp17la*, *Usp17lb*, *Usp17lc*, *Usp17ld*, and *Usp17le* by qPCR.

At baseline (0 h), the expression of other USP17L genes remained largely unchanged in both *Usp17la*-KO and *Usp17lb*-KO T cells, suggesting minimal compensatory expression under steady-state conditions. However, upon TCR stimulation (24 h), we observed dynamic changes indicative of potential compensatory mechanisms.

Specifically, USP17LA deficiency led to upregulation of *Usp17ld* and *Usp17le*, potentially reflecting a compensatory attempt to restore inhibitory signaling. Conversely, USP17LB deficiency resulted in upregulation of *Usp17la* and downregulation of *Usp17le*, suggesting a feedback mechanism that may fine-tune T cell activation. These results support the notion that while USP17L family members may function non-redundantly under resting conditions, they exhibit context-dependent regulatory interplay upon T cell activation, potentially serving as a compensatory network to maintain immune homeostasis.

These findings are included in supplementary figure (Fig EV5C, D), and the corresponding interpretation has been incorporated into the revised Discussion section, with newly added text highlighted for clarity.

Minor comments:

row 110-111 : The term "high homology" is wrong. Homology is qualitative (common evolutionary ancestry? yes/no). You probably mean "high sequence similarity".

Response: We thank you for the helpful correction. We agree that “homology” is a qualitative term referring to shared evolutionary ancestry, and that the phrase “high homology” is inappropriate in this context. Accordingly, we have revised the sentence in revised manuscript to use the more accurate term “high amino acid sequence similarity” to describe the high degree of nucleotide and amino acid identity among USP17L family members. We sincerely thank you again for your thoughtful review, which has greatly helped us improve and refine our manuscript.

Dear Prof. Zhang

Thank you for the submission of your revised manuscript to EMBO reports. Your manuscript was evaluated again by Referee #1. As you will see from the report copied below, the referee supports publication after some remaining minor concerns have been addressed. Please either provide further data supporting your conclusion on MAPK and NF- κ B or tone down the conclusions and modify the title accordingly.

From the editorial side, there are also a few things that we need before we can proceed with the official acceptance of your study.

- 1) Please reduce the number of keywords to 5.
 - 2) Data availability: this section should only refer to datasets generated in your study, i.e., the mass spectrometry proteomics. For this dataset we need an URL that resolves directly to the dataset. All other references to datasets you re-analysed should be moved to a distinct methods section, or referred to in the methods were appropriate.
 - 3) Data reference to E-MATB-6081 needs to be reformatted. A data reference must provide the database name, accession number/identifiers and a resolvable link to the landing page from which the data can be accessed at the end of the reference. Further instructions are available at <https://www.embopress.org/page/journal/14693178/authorguide#referencesformat> Ideally you would cite - in addition - the paper that generated the dataset.
 - 4) You cite (Zhang et al., 2019) as a data reference. The datasets produced in this study were not deposited in a public repository? I had a quick look and could not find information on data deposition. If the data were not deposited, please cite the paper as a normal citation, w/o "data ref". If the data were deposited, you could add a data reference to the datasets themselves.
 - 5) Please rename the Conflict of interests section to Disclosure and Competing Interests Statement.
 - 6) Studies involving mice:
 - Please state details of authority granting ethics approval (IRB or equivalent committee(s) and provide the reference number for approval in the methods section.
 - Please also describe the housing conditions in the methods.
 - 7) Regarding the Author Contributions, we now use CRediT to specify the contributions of each author in the journal submission system. Therefore, please remove the Author Contributions from the manuscript file and make sure that the author contributions in our online manuscript tracking system are correct and up-to-date. The information you specified in the system will be automatically retrieved and typeset into the article. You can enter additional information in the free text box provided, if you wish.
 - 8) We do not allow to base statements on "data not shown", which you currently state on page 5. Please include some data to bolster the statement.
 - 9) The Supplemental information section is not required and needs to be removed from the manuscript.
 - 10) We perform a routine image integrity check on all revised manuscripts. Doing so we noticed the following issue, that need to be resolved:

There are unmarked splice sites within Figure 7E (all blots) and Figure EV4J (ERK, p-IKKa/b). The figure needs to be presented clearly. As detailed in our Author guidelines, "all splice sites must be marked and based on data from a single experiment." - <https://www.embopress.org/image-integrity>
Please mark all splices:
Use a visible line or space (e.g., a thin white or black line) to indicate where the blot was spliced.
- Please also provide the source data for the entire Figure EV4J.
- 11) Performing a further spot check on the source data: Is the Western blot in the source data for Figure 7D, "western IP anti-Myc" the correct one? The bands look quite different.
 - 12) Our production/data editors have asked you to clarify several points in the figure legends (see below). Please incorporate these changes in the manuscript and return the revised file with tracked changes with your final manuscript submission.
- A) Statistical test information. Only p-values that are actually shown in the figure panel(s) should (and must) be defined in the legends, all others should be removed from (or added to) the legend. Moreover, we ask for the specification of exact p-values,

unless the p-values are small ($p < 0.0001$), which can be reported as inequalities:

1. Please note that the exact p values are not provided in the legends of figures 1B, 3A-D; 4E, J; 6D, F, 7F-H; EV3 J, EV5 A.

2. Please indicate the statistical test used for data analysis in the legend of figure 1A

B) Replicates and error bars:

3. Please note that information related to n is missing in the legend of figure 1A

- Figure 2C, D, G, H, N & Figure 4G & Figure EV2F please show the individual data points in addition to the mean and error bars.

- Figure 2A, E, K need scale bars.

13) Finally, EMBO Reports papers are accompanied online by

A) a short (1-2 sentences) summary of the findings and their significance,

B) 2-3 bullet points highlighting key results and

C) a schematic summary figure that provides a sketch of the major findings (not a data image).

Please provide the summary figure as a separate file in PNG or JPG format at a size of 550x300-600 pixels (width x height).

Please note that the size is rather small and that text needs to be readable at the final size. Please send us this information along with the revised manuscript.

With kind regards,

Martina Rembold, PhD

Senior Editor

EMBO reports

=====

Referee #1:

The revisions made by Zhang et al. greatly improve the manuscript quality, and have addressed all but one of my major concerns. I think this issue could be easily and quickly resolved with either a slight modification to Figure 6 (showing the quantification of the 3 westerns) or with a change to the title.

Response to the editor: I think that addressing reviewer 2's question #2 did not change the overall conclusions from the manuscript. The new information provided by the authors introduces directions for future mechanistic studies (e.g., how does USP17Lb regulate USB17La, and how does that change anti-tumor and inflammatory responses in T cells?). I think that the new data in EV5C-D fully address this concern.

Regarding Major critique #3:

I wrote: The title is misleading- only NFAT activation is impacted by USP17LA. MAPK and NFkB activity were not increased, and these should be removed from the title.

The authors respond:

Our data clearly demonstrated that USP17LA deficiency enhances two major TCR downstream signaling pathways-MAPK and NF- κ B-as evidenced by increased phosphorylation of ERK and P65, respectively. In parallel, USP17LA overexpression suppressed NFAT transcriptional activity, further supporting its regulatory role across multiple signaling axes. Specifically, in Fig 6A-D, we showed that phosphorylation of ERK and P65 was significantly increased in both primary T cells and EL4 cells following USP17LA deletion or knockdown, under anti-CD3/CD28 or PMA/ionomycin stimulation. In Fig 6E-F, the luciferase reporter assay in HEK293T cells demonstrated reduced NFAT activity upon USP17LA overexpression. Taken together, these findings support our conclusion that USP17LA negatively regulates T cell activation via coordinated modulation of the MAPK, NF- κ B, and NFAT pathways.

Remaining concern:

While the western blot in Figure 6B-C demonstrated an increase in phosphorylation of ERK and P65, the parallel approach in Fig 6E-F showed that MAPK, NF- κ B were unaffected by perturbation of USP17LA. It is not clear to me why the results from the western blot supersede the results from the reporter assay.

No data other than a single western blot is presented to substantiate the role of USP17LA in MAPK or NF- κ B. Therefore, having these elements in the title of the manuscript could result in improper amplification of this weakly supported conclusion.

My recommendation is that it should be removed from the title, or some additional information as to the reproducibility of the increase in phosphorylation of ERK and P65 should be included in the figure. One idea would be to quantify the band intensity from the 3 western blots.

Minor concerns:

Line 100- Figure 1B is inappropriately referenced

Figure 4 J- Where are the error bars, as stated in the legend. Are the bars too small to see?

Line 147-149 the data show decreased % of CD8+ CM cells, not increased, as written in the text.

Point-by-point response to the Editor and Reviewer:

We sincerely thank the Editor and Reviewer for their thoughtful and continued evaluation of our manuscript, and for recognizing the improvements made in the previous revision. We are very grateful for the constructive feedback received in this round, which has further strengthened the clarity, accuracy, and overall impact of our study. Below, we provide a detailed point-by-point response to all comments. All corresponding revisions have been incorporated into the manuscript and are highlighted in yellow.

Referee #1:

The revisions made by Zhang et al. greatly improve the manuscript quality, and have addressed all but one of my major concerns. I think this issue could be easily and quickly resolved with either a slight modification to Figure 6 (showing the quantification of the 3 westerns) or with a change to the title.

Regarding Major critique #3:

Response to the editor: I think that addressing reviewer 2's question #2 did not change the overall conclusions from the manuscript. The new information provided by the authors introduces directions for future mechanistic studies (e.g., how does USP17Lb regulate USB17La, and how does that change anti-tumor and inflammatory responses in T cells?). I think that the new data in EV5C-D fully address this concern.

I wrote: The title is misleading- only NFAT activation is impacted by USP17LA. MAPK and NFkB activity were not increased, and these should be removed from the title.

The authors respond:

Our data clearly demonstrated that USP17LA deficiency enhances two major TCR downstream signaling pathways-MAPK and NF- κ B-as evidenced by increased phosphorylation of ERK and P65, respectively. In parallel, USP17LA overexpression suppressed NFAT transcriptional activity, further supporting its regulatory role across multiple signaling axes. Specifically, in Fig 6A-D, we showed that phosphorylation of ERK and P65 was significantly increased in both primary T cells and EL4 cells following USP17LA deletion or knockdown, under anti-CD3/CD28 or PMA/ionomycin stimulation. In Fig 6E-F, the luciferase reporter assay in HEK293T cells demonstrated reduced NFAT activity upon USP17LA overexpression. Taken together, these findings support our conclusion that USP17LA negatively regulates T cell activation via coordinated modulation of the MAPK, NF- κ B, and NFAT pathways.

Remaining concern:

While the western blot in Figure 6B-C demonstrated an increase in

phosphorylation of ERK and P65, the parallel approach in Fig 6E-F showed that MAPK, NF- κ B were unaffected by perturbation of USP17LA. It is not clear to me why the results from the western blot supersede the results from the reporter assay.

No data other than a single western blot is presented to substantiate the role of USP17LA in MAPK or NF- κ B. Therefore, having these elements in the title of the manuscript could result in improper amplification of this weakly supported conclusion.

My recommendation is that it should be removed from the title, or some additional information as to the reproducibility of the increase in phosphorylation of ERK and P65 should be included in the figure. One idea would be to quantify the band intensity from the 3 western blots.

Response: We sincerely thank the reviewer for this valuable comment. In line with the reviewer's suggestion, we have revised the title to “The deubiquitinase USP17LA negatively regulates T-cell activation and attenuates anti-tumor immunity”, focusing on the broader conclusion while retaining detailed pathway evidence in the Results and Discussion.

Regarding the discrepancy between systems, USP17LA deletion consistently increased ERK and P65 phosphorylation in EL4 and primary T cells, whereas no changes in NF- κ B or AP-1 reporter activity were observed in HEK293T cells. We attribute this difference to the cellular context: T cells harbor intact TCR signaling cascades, while HEK293T cells lack this machinery. Supporting this, even the positive control DUSP2—previously reported to negatively regulate MAPK signaling in T cells (*Nat Immunol.* 2020, 21:287-297. **PMID:** 31932812)—failed to suppress AP-1 activity in HEK293T cells, highlighting the limitations of this system for assessing MAPK and NF- κ B pathways. In contrast, NFAT signaling, which relies on the calcium–calcineurin pathway preserved in HEK293T cells, was reliably reflected in the reporter assay.

Accordingly, we have moderated our interpretation, revised the title, and updated the Discussion section to clarify these points. We thank the reviewer again for the constructive feedback, which improved the clarity and balance of our conclusions.

Minor concerns:

Line 100- Figure 1B is inappropriately referenced

Response: We thank the reviewer for pointing out this mistake. We have corrected the reference; the intended figure is now accurately cited in the revised manuscript.

Figure 4 J- Where are the error bars, as stated in the legend. Are the bars too small to see?

Response: We appreciate the reviewer's attention to detail. The error bars are indeed present but are not readily discernible due to minimal variance among replicates. We have clarified this in the figure legend.

Line 147-149 the data show decreased % of CD8+ CM cells, not increased, as written in the text.

Response: We thank the reviewer for catching this discrepancy. The text has been corrected to state that the percentage of CD8⁺ central memory (CM) cells is decreased, in accordance with the data shown.

Editorial comments:

1. Keywords

→ Reduced to 5 keywords in the revised manuscript.

2. Data availability

→ Now restricted to datasets generated in this study (mass spectrometry proteomics).

→ Deposited to [ProteomeXchange/PRIDE accession and URL].

→ References to re-analyzed datasets have been moved to the Methods section.

3. Data reference formatting (E-MTAB-6081)

→ Reformatted according to EMBO style.

4. Data reference (Zhang et al., 2019)

→ Corrected: cited as a normal reference (not a data reference), as no dataset was deposited.

5. Conflict of Interests section

→ Renamed to *Disclosure and Competing Interests Statement*.

6. Ethics for animal studies

→ Methods updated to include:

- Name of the approving authority (IACUC).
- Approval reference number.
- Housing conditions.

7. Author Contributions

→ Removed from the manuscript file. Verified and updated in the online submission system using the CRediT taxonomy.

8. “Data not shown” (page 5)

→ Replaced with actual representative data, now included in the revised figures.

9. Supplemental Information section

→ Removed as requested.

10. Image integrity (splice sites in Fig. 7E and Fig. EV4J)

→ All splice sites clearly marked with thin lines.

→ Source data for the entire Fig. EV4J provided.

11. Western blot for Fig. 7D (IP anti-Myc)

→ Verified and replaced with the correct blot in the source data file. The previous blot was mistakenly from another replicate.

12. Figure legend clarifications

- Exact *p*-values provided for all relevant figures (1B, 3A–D, 4E, J, 6D, F, 7F–H, EV3J, EV5A).
- Statistical test specified in Fig. 1A legend.
- *n* values included in Fig. 1A legend.
- Individual data points shown in Fig. 2C, D, G, H, N, Fig. 4G, and Fig. EV2F.
- Scale bars added to Fig. 2A, E, K.

13. Summary and Highlights

→ Provided as requested:

- **Summary (1–2 sentences):** *USP17LA loss enhances T-cell function and anti-tumor ability, suggesting a potential target for cancer immunotherapy.*
- **Bullet points:**
 - *Usp17la-deficient T cells exhibit enhanced cytokine production, proliferative capacity, and anti-tumor activity.*
 - *USP17LA deubiquitinates RACK1 to negatively regulate NFAT signaling.*
 - *USP17LA represents a potential therapeutic target in cancer immunotherapy.*
- **Schematic summary figure:** Provided the schematic summary figure in two formats (550 × 300 pixels and 500 mm × 300 mm at 600 DPI) for the editor to

select the most suitable version.

We appreciate the opportunity to improve our work and hope the revised version meets the requirements for publication in *EMBO reports*.

Prof. Duanwu Zhang
Fudan university
131 Dong'an Road
Shanghai, Shanghai 200032
China

Dear Prof. Zhang,

I am very pleased to accept your manuscript for publication in the next available issue of EMBO reports. Thank you for your contribution to our journal.

Yours sincerely,
